# Single-Step Operator Learning for Conditioned Time-Series Diffusion Models

**Hui Chen**
Department of Statistics
University of Wisconsin - Madison
`hchen795@wisc.edu`

**Vikas Singh**
Department of Biostatistics and Medical Informatics
University of Wisconsin - Madison
`vsingh@biostat.wisc.edu`

## Abstract

Diffusion models have achieved significant success, yet their application to time series data, particularly with regard to efficient sampling, remains an active area of research. We describe an operator-learning approach for conditioned time-series diffusion models that gives efficient single-step generation by leveraging insights from the frequency-domain characteristics of both the time-series data and the diffusion process itself. The forward diffusion process induces a structured, frequency-dependent smoothing of the data's probability density function. However, this frequency smoothing is related (e.g., via likelihood function) to easily accessible frequency components of time-series data. This suggests that a module operating in the frequency space of the *time-series* can, potentially, more effectively learn to reverse the frequency-dependent smoothing of the *data distribution* induced by the diffusion process. We set up an operator learning task, based on frequency-aware building blocks, which satisfies semi-group properties, while exploiting the structure of time-series data. Evaluations on multiple datasets show that our single-step generation proposal achieves forecasting/imputation results comparable (or superior) to many multi-step diffusion schemes while significantly reducing inference costs. Our code is available at: `https://github.com/vsingh-group/SSOL-timeseries`.

## 1 Introduction

Generative modeling [Song and Ermon, 2019, Ho et al., 2020] has benefited immensely from the capabilities of modern diffusion models, which can capture complex data distributions across many different domains: from image synthesis to high-fidelity audio generation [Kong et al., 2021, Chen et al., 2021]. However, the application of these models to time-series data is still in its early stages [Tashiro et al., 2021, Rasul et al., 2021, Li et al., 2022, Alcaraz and Strodthoff, 2023, Yuan and Qiao, 2024, Fan et al., 2024, Li et al., 2025, Ye et al., 2025], and poses unique challenges. One widely acknowledged obstacle, which applies more generally to diffusion models, is the computational burden of the iterative denoising process [Song et al., 2021a, Salimans and Ho, 2022, Lu et al., 2022]. We need sequential evaluations during both training and inference, which makes the process slow and resource-intensive. A few of these issues are further compounded for time-series data with long time horizons. While excellent progress is being made to mitigate these difficulties broadly [Salimans and Ho, 2022, Lu et al., 2022], it is natural to ask whether the specific properties and the structure of time-series data may allow simplifications to diffusion-based generative modeling that retain the expressiveness/capacity of the models but can reduce the computational burden, especially for the sampling phase.

We seek to approach the problem above by viewing the forward diffusion process as operators [Li et al., 2021, Kovachki et al., 2023], where a suitably parameterized operator governs the evolution of data distributions across noise scales. This operator acts as a semigroup [Pazy, 1983, Henry, 1981], governing

how the data degrades, as noise is progressively introduced. We hypothesize that the characteristics of time-series data make them well-suited to benefit from this view. One observation is that the diffusion operator performs smoothing on the data's probability density function, systematically attenuating different frequency components at different rates, as shown in Fig. 1. This *frequency smoothing of the probability density* is distinct from but related to modulation of the frequency-space representations of the time-series data, and follows a predictable pattern. While frequency-dependent smoothing via diffusion occurs broadly, its structure is simpler to interpret and verify in time-series data, where frequency components directly correspond to

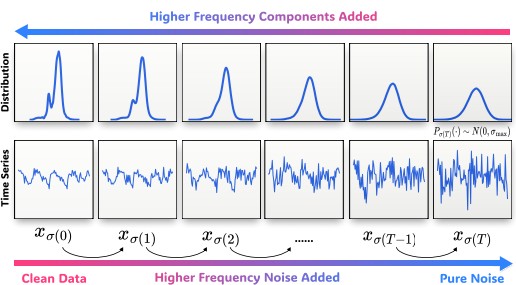

Figure 1: Dual view: forward diffusion smooths out the frequency distribution (top). Simultaneously, individual time-series trajectories gain high-frequency noise (bottom).

temporal patterns, such as daily or weekly cycles. This suggests that a scheme to characterize and leverage this behavior could potentially invert (or learn to invert) the diffusion process more efficiently. This is the specific direction we explore here, checking if rapid generation is possible while preserving the expressiveness that diffusion models are favored for.

Our overarching goal is to learn an approximation of the inverse of the diffusion operator, in a way that we can use it to effectively and efficiently reverse the noise addition process. Unlike conventional diffusion models that rely on iterative denoising, we seek to achieve this in a single step, reducing the compute cost. This capability is due to a specialized (but obtained with a relatively small set of adjustments) architecture that includes an operator trained to reconstruct and prioritize the distinct frequency components of the time-series. By conditioning this operator on both the starting and target noise levels, we can guide the model with a clear objective: to transform a noisy signal at a given scale to a cleaner representation at another scale, effectively traversing the trajectory defined by the diffusion semigroup. The training scheme enforces the composition property inherent to semigroups, ensuring that the learned transformations are consistent across different noise scales.

The main **contributions** of this paper are: **(a)** We introduce a new construction for single-step sampling in conditioned time-series diffusion models, by inverting the diffusion process. The architecture is based on operator learning principles and seeks to leverage the distinct roles of frequency components during denoising in the case of time-series data. **(b)** We validate the effectiveness of the idea on an extensive set of experiments where we match or exceed the performance of baselines while significantly reducing computational overhead.

## 2 Preliminaries: diffusion models

A continuous-time diffusion model [Song et al., 2021b] defines a forward process that gradually injects noise into data through a stochastic differential equation (SDE):

$$\mathrm{d}\boldsymbol{x} = f(\tau)\boldsymbol{x}\,\mathrm{d}\tau + g(\tau)\,\mathrm{d}\boldsymbol{w}_\tau, \tag{1}$$

where $\{\mathbf{x}(\tau)\}_{\tau=0}^T \subset \mathbb{R}^d$, $\mathbf{x}(\tau) = (x_0(\tau), x_1(\tau), \dots, x_{d-1}(\tau))^\top$ denotes the univariate time-series trajectory of length $d$ indexed by continuous diffusion time (or noise scale) $\tau \in [0, T]$, $\boldsymbol{w}_\tau$ is a standard $d$-dimensional Wiener process, $f(\tau)\boldsymbol{x}$ is a time-dependent linear drift in $\boldsymbol{x}$, and $g(\tau)$ is a time-dependent term to control the diffusion amplitude. This process transforms clean data $\boldsymbol{x}_0$ into increasingly noisy samples $\boldsymbol{x}(\tau)$. For notational simplicity, we write $\boldsymbol{x}_\tau = \boldsymbol{x}(\tau)$.

**Forward process.** The forward process progressively *obscures* information in the data distribution. The marginal distribution $p_\tau(\boldsymbol{x})$ of the noisy data $\boldsymbol{x}_\tau$ at noise scale $\tau$ can be expressed as a convolution of the original data, say $p_{\text{data}}(\boldsymbol{x})$ with a kernel. Specifically, the marginal distribution of the noised data $\boldsymbol{x}_\tau$ evolves according to:

$$p_{\sigma(\tau)}(\boldsymbol{x}) = \frac{1}{s(\tau)^d}\Big[p_{\text{data}} * \mathcal{N}\big(\boldsymbol{0}, \sigma^2(\tau)\,\mathbf{I}\big)\Big]\big(\tfrac{\boldsymbol{x}}{s(\tau)}\big), \tag{2}$$

where $p_{\text{data}}$ denotes the empirical data distribution of the "clean" data $\boldsymbol{x}_0$, $s(\tau) = \exp\big(\int_0^\tau f(u)\,\mathrm{d}u\big)$, and $\sigma^2(\tau) = \int_0^\tau \frac{g(u)^2}{s(u)^2}\,\mathrm{d}u$ are time-dependent scaling and variance functions respectively. The

convolution with increasingly fatter Gaussian kernels controlled by $\sigma^2(\tau)$ progressively suppresses high-frequency details by smoothing them out. The marginal distribution of $\boldsymbol{x}_\tau$ depends on $\tau$ through noise scale $\sigma(\tau)$, so we equivalently write $p_{\sigma(\tau)}(\boldsymbol{x}) = p_\tau(\boldsymbol{x})$. We denote $\sigma(\tau)$ by $\sigma$ or $\tau$ whenever its $\tau$-dependence is clear from context.

**Reverse process and loss.** To reverse this information loss and generate samples from the original data distribution, one often estimates the gradient $\nabla_{\boldsymbol{x}} \log p_\tau(\boldsymbol{x})$, which provides the local direction of the steepest ascent toward regions of higher likelihood under $p_\tau$. This is often modeled via a network that is trained to approximate the score function or equivalently to denoise the noisy data. The specific parameterization of the network can vary, with different choices leading to different reverse-time diffusion processes. One successful instance of this strategy, which we use in this work, is the EDM model in Karras et al. [2022]. In this approach, one trains a neural network $H_\theta$ to predict clean data from noisy samples $x_\sigma \sim \mathcal{N}(x, \sigma^2 \mathbf{I})$ using a skip-residual architecture:

$$H_\theta(\boldsymbol{x}_\sigma, \sigma) = C_{\text{skip}}(\sigma)\boldsymbol{x}_\sigma + C_{\text{out}}(\sigma)\left[B_\theta\left(C_{\text{in}}(\sigma)\boldsymbol{x}_\sigma, C_{\text{noise}}(\sigma)\right)\right], \tag{3}$$

where $B_\theta$ is the core network block (e.g., a U-net in images and wavenet in time-series) and $C_{\text{skip}}$, $C_{\text{out}}$, $C_{\text{in}}$, and $C_{\text{noise}}$ are scaling coefficients that depend on the noise level at $\sigma$. For example, $C_{\text{in}}$ and $C_{\text{out}}$ can perform scaling to ensure that the input and output have unit variance. The model is trained by minimizing the weighted denoising objective:

$$\mathcal{L}(\theta) = \mathbb{E}_{\sigma, \boldsymbol{\epsilon}, \boldsymbol{x}_0}\left[\omega(\sigma)|H_\theta(\boldsymbol{x}_0 + \sigma\boldsymbol{\epsilon}, \sigma) - \boldsymbol{x}_0|_2^2\right], \tag{4}$$

with noise levels $\sigma$ sampled from a log-normal distribution, where $\omega(\cdot)$ is a signal-to-noise ratio weight function [Karras et al., 2022], and $\boldsymbol{\epsilon}$ is a sample from a standard normal distribution.

# 3 Forward diffusion for time-series data

Reversing a diffusion process is difficult, especially in a single step. So, we need to leverage all available structural information about the data and the diffusion process itself. In particular, we will exploit the semigroup property of forward diffusion, which gives a compositional structure in how noise is added across time. Further, we want to use the distinct frequency-domain characteristics of time-series data and its link to the smoothing induced by the forward diffusion. We begin by examining how forward diffusion transforms the data distribution. This will yield some structural constraints we can leverage. Next, we delve into the frequency-domain implications of this forward process, demonstrating how the operator's action leads to a structured, frequency-dependent attenuation of information in the data's probability distribution which emphasizes the difficulty of the problem.

## 3.1 Time-inhomogeneous Markov evolution

We now analyze how the diffusion SDE (1) evolves the distribution of data over time (since our data is time-series, "time" is overloaded). By classical SDE theory [Øksendal, 2003], for each initial condition $\boldsymbol{x}_0$, there is a unique strong solution $\boldsymbol{x}_\tau$. Denoting the law (distribution) of $\boldsymbol{x}_\tau$ by $p_\tau$, we know the evolution of $p_\tau$ is governed by the forward Kolmogorov (Fokker–Planck) equation [Henry, 1981, Evans, 1998, Grafakos, 2014]:

$$\frac{\partial p_\tau}{\partial \tau} = L^*(\tau)\, p_\tau, \qquad p_{|\tau=0} \coloneqq p_{\text{data}}, \tag{5}$$

where the operator $L^*(\tau)$ captures both the drift and diffusion effects: $L^*(\tau)\, p = -\nabla \cdot \left[f(\tau)\, \boldsymbol{x}\, p\right] + \frac{1}{2}\, g(\tau)^2 \Delta p_\tau$, with $\Delta$ denoting the Laplacian in $\mathbb{R}^d$. We denote $p_{\text{data}}$ simply by $p_0$.

To capture how the density evolves from any time (noise scale) $\gamma$ up to a later time $\tau$, we can define a time-inhomogeneous Markov semigroup. Under mild regularity assumptions of globally Lipschitz drift ($f(\tau)\boldsymbol{x}$ is linear in $\boldsymbol{x}$) and uniformly nondegenerate diffusion coefficients ($g(\tau)^2$ bounded away from 0), classical SDE theory [Øksendal, 2003] guarantees pathwise existence and uniqueness of solutions to the forward diffusion process (1). Equivalently, the corresponding Fokker–Planck equation (5) is well-posed under these same conditions, which ensures the induced semigroup uniquely determines the transition distribution.

**Definition 3.1** (Two-parameter Markov family in $L^1$)**.** For each $0 \leq \gamma \leq \tau$, define the operator $S_{\gamma \to \tau}^* : L^1(\mathbb{R}^d) \to L^1(\mathbb{R}^d)$ to be the unique linear operator that sends an initial density $p_\gamma$ at time

$\gamma$ to the solution $p_\tau$ at time $\tau$ of the Fokker–Planck equation (5), i.e. $p_\tau = S^*_{\gamma \to \tau}(p_\gamma)$. We call $\{S^*_{\gamma \to \tau}\}_{0 \le \gamma \le \tau}$ the *two-parameter Markov family* generated by (5). The family satisfies the semigroup composition law for any $0 \le \rho \le \gamma \le \tau$:

$$S^*_{\rho \to \gamma} \circ S^*_{\gamma \to \tau} = S^*_{\rho \to \tau}, \quad S^*_{\tau \to \tau} = \text{Identity}.$$

Moreover, each $S^*_{\gamma \to \tau}$ is positivity- and mass-preserving.

**Relevance.** The semigroup property captures the Markov nature of the process – the evolution from time $\rho$ to $\tau$ can be *decomposed* into the evolution from $\rho$ to $\gamma$ followed by evolution from $\gamma$ to $\tau$. This Markov perspective will be useful later (Sec. 4) when we attempt to *invert* the forward diffusion with a reverse operator, where this property can offer constraints on estimating those primitives that will compose to give us longer-range transitions. However, recovering the original distribution is not simply applying the same semigroup in reverse, because diffusion is dissipative and thus not inherently invertible. We can see this by evaluating how extensive the dissipation is in practice.

## 3.2 Forward diffusion: exponential damping

We now check that the dissipative process leads to *exponential suppression of high-frequency details*. In essence, when the forward SDE spreads out and smooths the data distribution, it "dampens" or attenuates fine-scale modes. This behavior becomes clear by examining the distribution's Fourier transform. We formalize this in Theorem 3.2 below.

**Theorem 3.2** (Exponential decay and instant smoothing)**.** *Let $p_0$ be an initial probability density on $\mathbb{R}^d$, and assume that $p_\tau$ solves the forward Kolmogorov equation (5). Define the spatial Fourier transform $\widehat{p_\tau}(\boldsymbol{\xi}) = \int_{\mathbb{R}^d} e^{-i\,\boldsymbol{\xi} \cdot \boldsymbol{x}} p_\tau(\boldsymbol{x})\,d\boldsymbol{x},\ \boldsymbol{\xi} \in \mathbb{R}^d$. Then, for each fixed $\boldsymbol{\xi} \ne 0$ and $\tau > 0$, the quantity $\widehat{p_\tau}(\boldsymbol{\xi})$ is suppressed by a Gaussian factor in $\|\boldsymbol{\xi}\|^2$:*

$$\widehat{p_\tau}(\boldsymbol{\xi}) = \widehat{p_0}\big(s(\tau)\,\boldsymbol{\xi}\big)\, \exp\!\Big(-\tfrac{1}{2}\,\|\boldsymbol{\xi}\|^2\, s(\tau)^2\, \sigma^2(\tau)\Big), \tag{6}$$

*where $s(\tau) = \exp\!\big(\int_0^\tau f(u)\,du\big)$ and $\sigma^2(\tau) = \int_0^\tau \frac{g(u)^2}{s(u)^2}\,du$. Hence, each nonzero Fourier mode in $\hat{p}_\tau(\boldsymbol{\xi})$ decays exponentially in $\|\boldsymbol{\xi}\|$ for fixed time $\tau \ge 0$ and in $\tau$ for fixed $\|\boldsymbol{\xi}\| \ne 0$, provided that $s(\tau)^2 \sigma^2(\tau)$ grows in $\tau$.*

**Example 3.1.** Consider the simple SDE $dx = b\,dw_\tau,\ b > 0$, with solution $x_\tau = x_0 + b\,w_\tau$. Its distribution is normal with mean $x_0$ and variance $b^2\tau$, so the PDF spreads out as $\tau$ grows, flattening the peak and widening the tails.

This reflects the damping of high-frequency components (in terms of the data distribution). In other words, while the overall density becomes progressively smoother and more diffuse (losing high-frequency details), individual realizations exhibit increasingly large random fluctuations (for time-series data, this means introduction of high frequency noise). It is worth emphasizing that the smoothing of the PDF alongside increased noise in individual samples arises because we are looking at the distribution of possible trajectories, not the trajectory of any single sample path. Recognizing this dual behavior motivates a frequency-aware approach that exploits the information in the transform space of the time-series data, where we can analyze and manipulate the frequency components of the data itself. By understanding how the diffusion process affects the data distribution in the frequency domain, we can design our model to selectively restore and reweight dominant frequencies, more effectively reversing the smoothing effect on the PDF. This selective reweighting, if we can learn it, can help restore essential features of the initial distribution.

As one would expect, we notice that the same exponential damping of high-frequency components also manifests in *other* transform spaces, including wavelet spaces. In particular, we find that the solution $p_\tau(\cdot)$ to equation (5) exhibits an *instant smoothing* property: as soon as $\tau > 0$, $p_\tau$ belongs to *every* Sobolev space $H^\kappa(\mathbb{R}^d)$. This implies rapid decay of its coefficients in any standard multi-scale representation (e.g., wavelets), and so for other types of data, our frequency informed/aware module can be adjusted based on other transforms best suited for the data at hand.

**Corollary 3.3** (Instant smoothing)**.** *With the assumptions in Thm 3.2, assume additionally that $p_\tau \in L^2(\mathbb{R}^d)$ for each $\tau \ge 0$. Then for every $\tau > 0$ and for every real $\kappa \ge 0$, we have $p_\tau \in H^\kappa(\mathbb{R}^d)$. Equivalently, the wavelet coefficients $b_{j,k}(p_\tau)$ at scale $j$ position $k$ satisfy $\sum_{j,k} 2^{2j\kappa}\,\big|b_{j,k}(p_\tau)\big|^2 < \infty$ for each $\kappa \ge 0$. Hence $p_\tau$ is* infinitely differentiable *at any $\tau > 0$.*

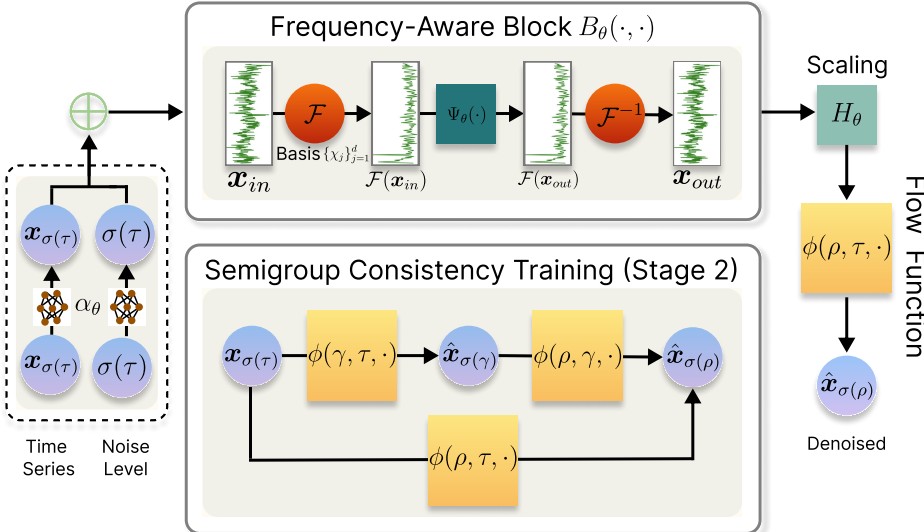

Figure 2: Overview of our single-step inverse operator for conditioned time-series diffusion models. A noisy input $\mathrm{x}_{\sigma(\tau)}$ is passed through a frequency-aware block to selectively restore high-frequency components. A single-step operator $\phi$ then enforces the semigroup property to denoise across noise levels in a single step. The bottom block illustrates the semigroup consistency training that enables this single-step capability by enforcing $\phi(\gamma, \tau, \cdot) \circ \phi(\rho, \gamma, \cdot) = \phi(\rho, \tau, \cdot)$.

**Relevance.** The above results show a clear picture of forward diffusion: **(i)** it forms a Markov semigroup $S^*_{\gamma \to \tau}$ that is not trivially invertible, and **(ii)** it exponentially suppresses high-frequency details in both Fourier and wavelet (and other) domains. To recover such lost details, standard diffusion models resort to many small reverse steps. We will seek to restore these details *in one step* by exploiting information regarding the frequency domain of the time-series described above and the composition property, in conjunction.

## 4 Single-step inverse operator

The discussion suggests that we need two ingredients: **(i)** a *frequency-aware* block that uses the frequency-domain characterization of the time-series to restore the damped high-frequency components (of the data distribution), and **(ii)** a semigroup-based reverse operator that is well-defined across different time intervals. We begin by describing how the block selectively boosts suppressed modes in § 4.2, then show how to embed it within a flow operator $\phi$ that approximates $(S^*_{\gamma \to \tau})^{-1}$ from § 3.1.

### 4.1 Use case: time series completion

We will use a time-series completion task as an umbrella example to describe the components of our construction. Let $\boldsymbol{x} \in \mathbb{R}^{C \times d}$ be a multivariate time series with $C$ channels over $d$ discrete timesteps, and let $\mathbf{M} \in \{0, 1\}^{C \times d}$ be a binary mask indicating observed versus unobserved entries. We then write $\boldsymbol{x}_{\text{obs}} = \mathbf{M} \odot \boldsymbol{x}$ and $\boldsymbol{x}_{\text{target}} = (1 - \mathbf{M}) \odot \boldsymbol{x}$, where $\odot$ denotes elementwise multiplication. This formulation unifies various data completion tasks: *forecasting* arises when $\mathbf{M}$ masks future timesteps, while *imputation* arises when $\mathbf{M}$ zeroes out scattered entries within $\boldsymbol{x}$.

Our noisy sample $\boldsymbol{x}_\sigma$ is generated by injecting noise at level $\sigma$, e.g., $\boldsymbol{x}_\sigma = \boldsymbol{x} + \sigma \boldsymbol{\epsilon}$ with $\boldsymbol{\epsilon} \sim \mathcal{N}(\mathbf{0}, \mathbf{I})$. Our goal is to learn an *inverse* mapping $\phi$ that approximately recovers $\boldsymbol{x}$ (or its unobserved portion) from $\boldsymbol{x}_\sigma$ in a *single step* when conditioned on the observations $\boldsymbol{x}_{obs}$. To achieve this, §4.2 describes a *Frequency-Aware Block* that adaptively smoothes out unplausible frequency components and reweights dominant components operating in the frequency domain of the *data* (time-series). Then, in §4.3, we construct a single-step operator $\phi$ that enforces the semigroup properties of the diffusion process, ensuring coherent reconstructions under varying noise levels.

## 4.2 Design of frequency-aware block (FAB)

Let $\Psi_\theta(\cdot)$ be a neural embedding function (a small stack of residual layers) for spectral coefficients derived from a noisy input time series $\boldsymbol{x}_\tau$. We first project the input onto a chosen spectral basis $\{\chi_j\}$ to obtain coefficients $\langle \boldsymbol{x}_\tau, \chi_j \rangle$. This spectral basis is chosen to align with the inherent frequency-domain structure of time-series data. We use localized time-frequency transforms such as wavelet transform in practice, as it works well at capturing time-localized patterns. A Fourier basis decomposes signals into their constituent frequencies globally, while a wavelet basis provides localized time-frequency representation. The specific choice will depend on the time series at hand, considering its dynamic or piecewise-stationary frequency content (e.g., seasonal patterns, transient bursts). Thus, a localized time-frequency transform allows our model to adaptively emphasize different frequency bands at specific time points (here, time refers to timesteps in the time-series). A learnable modulation function $\alpha_\theta(\sigma(\tau))$ (a MLP with positional embeddings) scales these coefficients based on the noise level $\sigma(\tau)$. We define the block simply as:

$$B_\theta\big(\boldsymbol{x}_\tau, \sigma(\tau)\big) \;=\; \sum_j \Psi_\theta\bigg( \alpha_\theta\big(\sigma(\tau)\big) \cdot \big\langle \boldsymbol{x}_\tau, \chi_j \big\rangle \bigg) \chi_j. \tag{7}$$

Recall that the forward diffusion process progressively attenuates/smooths the high frequency components of the data's PDF. This block provides a signal to *counteract* this effect by operating in the frequency domain of the time-series data itself. The projection onto the spectral basis decomposes the time-series into its constituent frequency components. The learnable modulation function adjusts the amplitude of each data frequency component based on the noise level since the degree of smoothing in the PDF depends on $\tau$ (and so, $\sigma(\tau)$). At high noise levels, we must learn to use the higher data frequencies, recognizing that this is related to the smoothing of the PDF. This block helps associate how the attenuation pattern in the PDF space (due to diffusion) corresponds to the observed frequency content in the noisy data. Next, in §4.3, we integrate FAB into our single-step operator to enable the direct recovery of clean observed time series from noisy samples.

## 4.3 Design of inverse semigroup operators

We will now leverage the semigroup property. While conventional diffusion-based samplers approximate $(S_{\gamma \to \tau}^*)^{-1}$ via many small reverse steps, we describe a learnable *inverse* semigroup operator $\phi$ that reconstructs these lost components in a single step. This approach directly maps a noisy observation $x_{\sigma(\tau)}$ at noise level $\sigma(\tau)$ to a cleaner sample $x_{\sigma(\gamma)}$ at a lower noise level $\sigma(\gamma)$, where $\gamma < \tau$, guided by the frequency-domain information, while preserving the algebraic properties of the forward semigroup.

**Flow operator construction.** To ensure consistency with the semigroup structure, our learnable flow operator $\phi(\gamma, \tau, \cdot)$ is a time-dependent convex combination:

$$\phi(\gamma, \tau, \boldsymbol{x}) \;=\; \frac{\tau - \gamma}{\tau}\, H_\theta(\boldsymbol{x}) \;+\; \frac{\gamma}{\tau}\, \boldsymbol{x}, \tag{8}$$

where $H_\theta(\cdot)$ is a neural network that incorporates the FAB block $B_\theta$ in §4.2 internally via (3) and is conditioned on the noise level $\sigma(\tau)$. The linear interpolation ensures that the identity $\phi(\tau, \tau, \boldsymbol{x}) = \boldsymbol{x}$ holds (no denoising), maintaining consistency with the forward semigroup at zero step size (i.e., no noise added). Intuitively, $H_\theta(\cdot)$ attempts to denoise based on the amount of denoising needed, and leverages the FAB block to enable frequency-specific re-weighting and reconstruction of the fine-scale features lost by the smoothing effect of $S_{\gamma \to \tau}^*$.

**Two-stage training objective.** We train $\phi$ through a two-stage process that enforces both endpoint accuracy and global consistency with the semigroup property: **(i)** *Boundary/Endpoint.* We ensure that mapping from noise level $\tau$ to 0 does recover the clean sample: $\phi(0, \tau, \boldsymbol{x}_{\sigma(\tau)}) = \boldsymbol{x}_0$. This translates to the denoising loss in (4), where we sample $\sigma(\tau)$ from a log-normal distribution and $\boldsymbol{x}_{\sigma(\tau)} = \boldsymbol{x}_0 + \sigma(\tau)\boldsymbol{\epsilon}$. This objective aligns the single-step operator with standard diffusion model training at the boundary $\gamma = 0$, preventing trivial solutions or model collapse. **(ii)** *Semigroup composition property.* Next, we ensure $\phi$ approximates $(S_{\gamma \to \tau}^*)^{-1}$ globally by enforcing the composition property:

$$\phi(\rho, \gamma, \; \phi(\gamma, \tau, \boldsymbol{x})) \;=\; \phi(\rho, \tau, \boldsymbol{x}), \qquad \forall\, 0 \le \rho \le \gamma \le \tau \tag{9}$$

This enforces semigroup coherence: two single-step moves from $\tau \to \gamma$ and $\gamma \to \rho$ should match a single move from $\tau \to \rho$. To facilitate training, we use a linear adaptive schedule $\boldsymbol{N}(\cdot)$ that

progressively refines the temporal discretization grid as training proceeds, allowing $\phi$ to learn multi-step inversions. We begin with larger intervals between noise levels to encourage coarser approximation initially, then refine toward smaller intervals; directly starting with small intervals could trap the model in local minima by overtly focusing on fine-scale noise differences. Alg. 1 and Alg. 2 in Appendix B show the complete training and sampling procedures.

## 5 Experiments

In this section, we present our experimental findings. Our evaluation protocol consists of two stages: **(i)** assessing the quality of multivariate time series (MTS) completion on both forecasting and imputation tasks with conditions on a variety of datasets, and **(ii)** conducting an ablation study to analyze the effectiveness of our model components.

Table 1: Performance comparison of probabilistic diffusion models for conditioned time series generation (forecasting). Deterministic point forecasting methods are included as reference baselines to contextualize current forecasting capabilities. **Bold** values indicate best performance per dataset, while underlined values represent second-best results. ($\downarrow$).

| Model | Method | ETTm1 | | ETTm2 | | Weather | | Solar-Energy | | Electricity | | Traffic | |
|---|---|---|---|---|---|---|---|---|---|---|---|---|---|
| | | MSE | MAE | MSE | MAE | MSE | MAE | MSE | MAE | MSE | MAE | MSE | MAE |
| Point forecasting | NSformer ([2022]) | 0.440 | 0.430 | 0.277 | 0.343 | 0.226 | 0.270 | 0.266 | 0.270 | 0.191 | 0.295 | 0.653 | 0.360 |
| | TimesNet ([2023]) | 0.374 | 0.387 | 0.249 | 0.309 | 0.219 | 0.261 | 0.296 | 0.318 | 0.184 | 0.289 | 0.617 | 0.336 |
| | DLinear ([2023]) | 0.380 | 0.389 | 0.284 | 0.362 | 0.237 | 0.296 | 0.320 | 0.398 | 0.196 | 0.285 | 0.598 | 0.370 |
| | PatchTST ([2023]) | 0.370 | 0.390 | 0.251 | 0.312 | 0.223 | 0.258 | 0.259 | 0.321 | 0.205 | 0.307 | 0.463 | 0.311 |
| | SparseVQ ([2024]) | **0.363** | **0.380** | 0.242 | **0.302** | 0.225 | 0.256 | 0.256 | 0.286 | 0.182 | 0.267 | 0.480 | 0.300 |
| | iTransformer ([2024]) | 0.377 | 0.391 | 0.250 | 0.309 | 0.221 | **0.254** | 0.233 | 0.261 | 0.164 | 0.255 | **0.418** | 0.284 |
| Probabilistic forecasting | TimeGrad ([2021]) | 1.716 | 1.057 | 1.385 | 0.732 | 0.885 | 0.551 | 1.211 | 1.004 | 0.645 | 0.723 | 0.932 | 0.807 |
| | CSDI ([2021]) | 0.867 | 0.690 | 1.291 | 0.576 | 0.842 | 0.523 | 0.848 | 0.818 | 0.553 | 0.795 | 0.921 | 0.678 |
| | TimeDiff ([2023]) | 0.796 | 0.577 | 0.284 | 0.342 | 0.277 | 0.331 | 1.169 | 0.936 | 0.730 | 0.690 | 1.465 | 0.851 |
| | DiffusionTS ([2024]) | 1.030 | 0.744 | 2.372 | 1.232 | 0.563 | 0.574 | 0.749 | 0.740 | 1.072 | 0.856 | 1.473 | 0.815 |
| | TMDM ([2024]) | 0.607 | 0.558 | 0.524 | 0.493 | 0.244 | 0.286 | 0.295 | 0.317 | 0.222 | 0.329 | 0.721 | 0.411 |
| | D3U ([2025]) | 0.368 | 0.387 | 0.241 | **0.302** | 0.222 | 0.264 | 0.237 | 0.270 | 0.179 | 0.267 | 0.468 | 0.299 |
| | NsDiff ([2025]) | 0.488 | 0.455 | 0.281 | 0.352 | 0.248 | 0.293 | 0.242 | 0.307 | 0.209 | 0.306 | 0.637 | 0.373 |
| | SSOL (Ours) | 0.369 | 0.384 | **0.238** | 0.314 | **0.209** | 0.255 | **0.189** | **0.223** | **0.155** | **0.241** | 0.550 | **0.259** |

### 5.1 Setup: datasets and baselines

**Baselines.** We benchmark our method against two groups of baselines **(a)** *diffusion-based probabilistic models*: NsDiff [Ye et al., 2025], D3U [Li et al., 2025], TMDM [Li et al., 2024], Diffusion-TS [Yuan and Qiao, 2024], TimeDiff [Shen and Kwok, 2023], CSDI [Tashiro et al., 2021], TimeGrad [Rasul et al., 2021], DiffWave [Kong et al., 2021]; **(b)** *deterministic models*: iTransformer [Liu et al., 2024], PatchTST [Nie et al., 2023], DLinear [Zeng et al., 2023], TimesNet [Wu et al., 2023], NSformer [Liu et al., 2022] SparseVQ [Zhao et al., 2024], Forecasting and imputation experiments use task-specific subsets for appropriate comparisons. More information is provided in Appendix C.

**Datasets.** We evaluate our model on six widely used benchmarks: ETTh1, ETTm1 [Zhou et al., 2021], Weather, Electricity, Traffic [Wu et al., 2021], and Solar-Energy [Lai et al., 2017]. For forecasting, we mainly follow the experimental configurations in [Wu et al., 2023], including the same data processing and splitting protocol. All experiments use a 192-step prediction horizon, adopting default lookbacks for baselines that report at this horizon, or fixing lookback at 96 otherwise for fair comparison. For imputation, we fix the window length to 48, following [Yuan and Qiao, 2024].

### 5.2 Evaluations on MTS completion tasks

**Forecasting.** Forecasting results from our experiments are summarized in Table 1, with detailed probabilistic evaluation using the Continuous Ranked Probability Score (CRPS) provided in Table 2. Our method demonstrates *robust* performance across these diverse benchmarks, *matching or surpassing* diffusion-based probabilistic baselines. To give context for our method's probabilistic performance, we include comparisons with deterministic Transformer-based forecasting models (e.g., PatchTST, iTransformer). Although deterministic forecasting is not our primary focus, our approach remains *competitive* with these strong deterministic forecasters, frequently attaining top or near-top results. ***Summary.*** Overall, these findings suggest that our single-step design can handle complex temporal dependencies effectively in a 192-horizon prediction setting.

Table 2: Performance evaluation of probabilistic forecasting models using CRPS and CRPS$_{sum}$. **Bold** values indicate best performance per dataset, while underlined values represent second-best results.

| Model | Dataset Method | ETTm1 | | ETTm2 | | Weather | | Solar-Energy | | Electricity | | Traffic | |
|---|---|---|---|---|---|---|---|---|---|---|---|---|---|
| | | CRPS | CRPS$_{sum}$ | CRPS | CRPS$_{sum}$ | CRPS | CRPS$_{sum}$ | CRPS | CRPS$_{sum}$ | CRPS | CRPS$_{sum}$ | CRPS | CRPS$_{sum}$ |
| Probabilistic Forecasting | TimeGrad (2021) | 0.665 | 0.996 | 0.785 | 1.051 | 0.482 | 0.503 | 0.783 | 1.167 | 0.503 | 1.452 | 0.657 | 1.683 |
| | CSDI (2021) | 0.773 | 0.852 | 0.625 | 0.782 | 0.508 | 0.465 | 0.649 | 0.681 | 0.465 | 0.823 | 0.612 | 1.275 |
| | TimeDiff (2023) | 0.454 | 0.846 | 0.316 | 0.180 | 0.293 | 0.400 | 0.900 | 1.164 | 0.475 | 0.594 | 0.671 | 0.823 |
| | TMDM (2024) | 0.429 | 0.633 | 0.380 | 0.226 | 0.226 | 0.292 | 0.375 | 0.267 | 0.446 | 0.137 | 0.552 | 0.179 |
| | D3U (2025) | **0.285** | 0.749 | 0.243 | 0.141 | **0.207** | **0.283** | **0.186** | 0.266 | **0.202** | 0.160 | **0.232** | 0.186 |
| | NsDiff (2025) | 0.350 | 1.614 | 0.256 | 1.315 | 0.244 | 1.873 | 0.300 | 27.64 | 0.290 | 29.65 | 0.378 | 119.4 |
| | SSOL (Ours) | 0.376 | **0.553** | **0.184** | **0.131** | 0.348 | 0.384 | 0.211 | **0.195** | 0.224 | **0.116** | 0.268 | **0.176** |

**Imputation.** We follow the same geometric-mask strategy setup as in Zerveas et al. [2021], Yuan and Qiao [2024], where each missing segment's length is sampled from a geometric distribution, ensuring that the data is masked in consecutive segments rather than at random individual points. This setup aligns with real-world sensing scenarios, such as sensor failures or intermittent transmission losses, where data typically drops out in bursts. We evaluate our approach on the ETTh1 and Energy datasets under missing ratios ranging from $10\%$ to $90\%$. In Figure 3, we observe that our proposed method consistently achieves a lower MSE than all baselines throughout the entire range of missing ratios. In Figure 5 in Appendix C, we show single-channel imputation examples.

**How can single-step outperform diffusion-based baselines?** Our algorithm is *efficient yet powerful* and only needs a single denoising step (NFE = 1) while consistently matching or outperforming multi-step (e.g., NFE = 20) diffusion-based approaches on all tested datasets. Conventional methods often rely on multi-step backward diffusion, where even moderate time increments can be ill-conditioned due to the exponential decay of high-frequency modes in the forward process (discussed in Thm 3.2). This necessitates many small reverse steps and fine-tuning of step sizes or noise schedules to mitigate error accumulation. In contrast, we learn a *global* operator. Of course, the *composition property* in (9) over all noise intervals is essential: a direct jump from one noise level to another must match the outcome of traversing any intermediate levels. This ensures the operator remains well-defined for partial intervals and helps stable one-shot reconstruction of fine-scale details, helped by the FAB block. Our model avoids the iterative error propagation and hyperparameter tuning (e.g., step sizes or scheduling) common in multi-step diffusion. Our end-to-end training over the full noise range yields faster inference and lower forecasting errors.

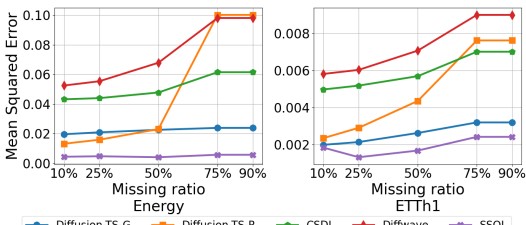

Figure 3: Imputation results. MSE of different imputation models under various missing ratios for the ETTh1 (left) and Energy (right) datasets.

## 5.3 Digging deeper into the operating profile of the model

**Ablation study.** We evaluate the impact of key components in our single-step conditioned generation model through systematic ablation experiments on forecasting benchmarks. All experiments maintain identical training configurations while selectively removing specific components: **(i)** *Complete model*: Full framework with composition property and FAB; **(ii)** *No composition property*: Omits composition training stage; **(iii)** *No frequency-aware block*: Uses standard residual block instead of FAB.

**Observations. (a)** *Composition.* The ablation results in Table 3 show that removing the composition property consistently increases both MSE and MAE across all benchmarks. This is because the composition property enforces the semigroup structure, ensuring stable transitions from high to low noise levels, even with large step sizes. Otherwise, the model degenerates into a direct mapping from high noise to clean data, and so cannot handle intermediate noise transitions effectively. **(b)** *FAB block:* The frequency-aware block is equally useful for model performance. While forward diffusion inherently suppresses high-frequency modes in the probability density function, our block's sample-level frequency adjustments enable the cumulative restoration of these modes at the distribution level. The increased error rates observed when removing this block, as shown in Table 3, show that precise per-sample frequency modulation is a meaningful signal.

Table 3: (a) Ablation study: impact of composition property and frequency-aware block removal. (b) Runtime gains due to single-step generation. ($\downarrow$).

**(a) Ablation study (Pred. Len = 96)**

| Models/Metric | ETTh1 MSE | ETTh1 MAE | ETTm1 MSE | ETTm1 MAE | Weather MSE | Weather MAE |
|---|---|---|---|---|---|---|
| w/o Composition | 0.402 | 0.433 | 0.362 | 0.394 | 0.190 | 0.268 |
| w/o FAB | 0.406 | 0.424 | 0.336 | 0.366 | 0.178 | 0.221 |
| Complete Model | **0.375** | **0.396** | **0.325** | **0.352** | **0.153** | **0.209** |

**(b) Runtime gains (ETTh1)**

| Model | Approach | Steps | Pred. Len = 96 MSE | MAE | CRPS | CRPS$_{sum}$ | Pred. Len = 192 MSE | MAE | CRPS | CRPS$_{sum}$ |
|---|---|---|---|---|---|---|---|---|---|---|
| NsDiff [2025] | Original | 20 | 0.543 | 0.496 | 0.371 | 1.539 | 0.603 | 0.525 | 0.394 | 1.638 |
|  | + SSOL | **1 (20×)** | 0.527 | 0.484 | 0.364 | 1.477 | 0.580 | 0.522 | 0.389 | 1.682 |
| D3U [2025] | Original | 20 | 0.415 | 0.426 | 0.317 | 0.699 | 0.471 | 0.463 | 0.340 | 0.799 |
|  | + SSOL | **1 (20×)** | 0.400 | 0.412 | 0.369 | 0.813 | 0.440 | 0.439 | 0.381 | 0.823 |
| SSOL | Reference | 1 | **0.375** | **0.496** | 0.377 | **0.564** | **0.421** | **0.422** | 0.405 | **0.600** |

**Efficiency.** As shown in Table 3, we achieve strong efficiency gains by reducing the sampling steps to only one (NFE = 1) – roughly $20\times$ improvement. When applied to recent diffusion models NsDiff and D3U, ours consistently improves prediction accuracy, reducing MSE and MAE in both 96 and 192-step forecasts while maintaining comparable CRPS metrics.

**Runtime analysis.** The additional runtime analysis table is shown in Appendix C.1. We provide actual training and inference times measured on identical hardware to address efficiency claims in Table 8. We used a computationally efficient variant of our method with reduced model size and shallower residual layers, while preserving the core frequency-aware block and semigroup composition training methodology. Our two-stage training exhibits comparable per-iteration times to baseline methods, with Stage 1 (boundary denoising) and Stage 2 (semigroup constraints) adding minimal overhead compared to standard single-stage training. Although we do not achieve a pure $20\times$ speedup in wall-clock time due to backbone architecture differences across methods, our approach shows strong efficiency gains during inference. The key findings are: **(a)** inference speedup of $3.5\times$ faster than D3U and $25\times$ faster than NsDiff, **(b)** memory efficiency with $35\%$ less GPU usage than D3U and $85\%$ less than NsDiff, and **(c)** performance parity, where, despite far fewer denoising steps, SSOL achieves comparable/better MSE/MAE. The efficiency gains come from our single-step design, while the additional training time for semigroup consistency is negligible compared to the inference savings.

**Are single-step outputs based on actual denoising?** In the single-step framework, one concern is whether the model can collapse to a trivial solution simply by ignoring the injected noise and behaving like a conventional regressor in the conditioning window. In that case, drawing multiple noisy samples would not meaningfully change the model's predictions, since each forward pass would converge to the same deterministic output. To check this, we conducted an experiment in which we varied the number of generated samples $n$ (see panel 3 of Fig. 4). We observe that as $n$ increases, the median of these $n$ samples consistently lowers MSE and MAE, indicating the model *does* react to each injected noise draw, i.e., it is genuinely denoising from different noisy realizations. If the model were ignoring noise, its outputs (and thus errors) would not improve with additional samples, suggesting that it learns the conditional distribution over possible future trajectories.

**Impact of shifted noise distribution.** We also tested conditional forecast generation with a much smaller $\sigma_{\max}$ at inference time, which produces a narrower prior distribution, creating a mismatch in the types of perturbations the model must denoise. The right panel of Figure 4 shows that the distribution shift degrades performance, indicating that the single-step operator depends on access to a broader range of noise levels during inference to capture the distribution's variability. This behavior is expected: our operator is trained to map from high noise levels, where forward diffusion has sufficiently smoothed the data distribution (in Thm 3.2), but reducing $\sigma_{\max}$ at inference violates this assumption by starting from insufficiently noisy priors. The degradation confirms our model performs diffusion-based generation and is not learning shortcuts that ignore the noise structure.

**Does single-step denoising recover the high-frequency components in distribution?** We can ask if our model can recover the fine-grained, high-frequency details via the frequency-aware block. In Fig. 4, the t-SNE plot (left) shows close proximity in the embedded space, suggesting that we can cover the overall data manifold in the forecasting setting. Further, the power spectral density curves in the second-left panel show that the generated distribution captures high-frequency components that align closely with the real data, indicating that FAB can denoise the noise in individual observations and re-injects the fine-scale details in the conditional distribution.

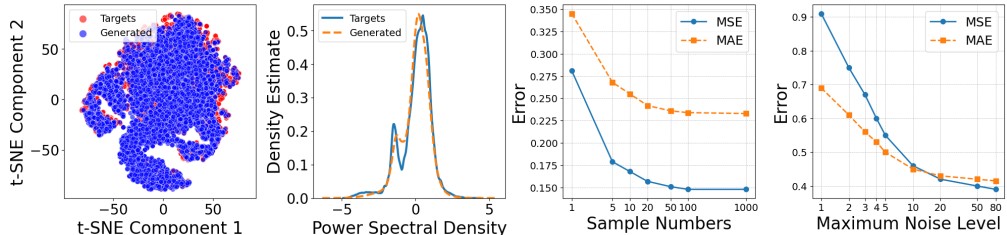

Figure 4: Left: t-SNE manifold comparison and power spectral density recovery on ETTm1. Right: Error analysis of conditioned generation on Electricity and ETTh1.

## 6 Related Work

**Single-step denoising.** Most diffusion models rely on multiple iterative denoising steps in the reverse process [Song and Ermon, 2019, Ho et al., 2020], which can be expensive at inference time. Recent research has therefore explored strategies for reducing the number of sampling steps. For instance, Nichol and Dhariwal [2021] propose a strided sampling schedule to skip certain reverse-time intervals, while Salimans and Ho [2022] proposes progressive distillation that halves the diffusion steps in each iteration through a teacher-student paradigm. More recently, Song et al. [2023] learn a direct mapping from any intermediate noisy point to the clean signal in a single step via consistency distillation. Similar single-step approaches have been developed for image generation [Frans et al., 2025, Geng et al., 2025], though these require domain-specific adaptations for time-series applications. Our work leverages classical principles in Pazy [1983], Henry [1981] to define a single-step inverse operator.

**Diffusion models for conditioned time series generation.** Time-series forecasting has evolved from classical statistical models such as ARIMA [Box and Jenkins, 1994] to advanced Transformer-based approaches [Liu et al., 2024, Nie et al., 2023] and more recently, diffusion based approaches. Proposals like Diffusion-TS [Yuan and Qiao, 2024], MG-TSD [Fan et al., 2024], TSDiff [Kollovieh et al., 2023], SSSD [Alcaraz and Strodthoff, 2023], D3VAE [Li et al., 2022], TimeGrad [Rasul et al., 2021], CSDI [Tashiro et al., 2021], and ImagenTime [Naiman et al., 2024] leverage diffusion-based generative modeling in interesting ways for probabilistic forecasting/imputation. Some recent results have run diffusion in the frequency domain: Crabbé et al. [2024] performs score matching on mirrored Brownian motion (in Fourier basis) and improves sample fidelity, and FIDE [Galib et al., 2024] inflates high-frequency coefficients and conditions on block maxima to preserve extreme events. However, these approaches still rely on multi-step sampling, efficient single-step generation for time-series diffusion models is largely unaddressed in these works.

## 7 Conclusions

We introduced a novel single-step operator learning approach for time-series diffusion models that enables efficient conditioned generation while maintaining high-quality results. By using insights from the frequency-domain characteristics of both time-series data and the diffusion process itself, our model can effectively reverse the smoothing effects of forward diffusion in one step. Extensive experiments across multiple datasets shows that our approach achieves comparable or better performance than existing multi-step diffusion and deterministic methods, while reducing computational costs quite drastically, which can make diffusion models more practical and efficient for numerous time-series applications.

## Acknowledgments and Disclosure of Funding

The authors are grateful to Sourav Pal, WonHwa Kim, Melanie Boly, and Giulio Tononi for discussions and input. Partial support for this work came from a contract to UW-Madison under the DARPA Strengthen program.

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

# A  Attenuation of high-frequency modes

A key observation of forward diffusion processes is the attenuation of high-frequency modes. In the non-degenerate linear drift diffusion setting (1), the associated Fokker–Planck operator (5) enforces exponential damping of the Fourier transform of the density at large wavenumbers. In the following, we provide detailed proofs of these properties (which are also stated in Sect. 3.2), then connect them to smoothing in Sobolev and wavelet spaces.

**Assumptions.**  Throughout this section, we fix measurable coefficient functions $f, g : [0, \infty) \to \mathbb{R}$ with $g(\tau) \geq \delta > 0$ and work under the following hypotheses:

(A1)  Initial data. The starting density satisfies $p_0 \in L^1(\mathbb{R}^d) \cap L^2(\mathbb{R}^d)$.

(A2)  Coefficient regularity.  $f, g \in L^1_{\mathrm{loc}}([0, \infty))$, ensuring that all integrals $\int_0^\tau f(u)\, du$ and $\int_0^\tau g(u)^2\, du$ are finite for every $\tau > 0$.

(A3)  Existence of an $L^1 \cap L^2$ density. For each $\tau > 0$, the Fokker-Planck in equation (5)

$$\partial_\tau p_\tau = \tfrac{1}{2}\, g(\tau)^2 \Delta p_\tau - \nabla \cdot \big[ f(\tau)\, \boldsymbol{x}\, p_\tau \big], \quad p_{|\tau=0} = p_0,$$

admits a solution $p_\tau \in L^1(\mathbb{R}^d) \cap L^2(\mathbb{R}^d)$ with sufficient decay at spatial infinity to justify term-wise Fourier transforms and integration by parts.

## A.1  Exponential damping in the Fourier domain

To analyze the high-frequency behavior, let us take the spatial Fourier transform of the probability density $p_\tau(\boldsymbol{x})$. Define

$$\widehat{p_\tau}(\boldsymbol{\xi}) \;=\; \int_{\mathbb{R}^d} e^{-i\, \boldsymbol{\xi} \cdot \boldsymbol{x}}\, p_\tau(\boldsymbol{x})\, \mathrm{d}\boldsymbol{x}, \quad \boldsymbol{\xi} \in \mathbb{R}^d.$$

The following theorem shows that for each fixed $\boldsymbol{\xi} \neq 0$, these modes decay at an exponential rate determined by the diffusion amplitude.

**Theorem A.1** (Exponential decay of Fourier modes). *Assume* (A1)–(A3) *hold. Put*

$$s(\tau) \;:=\; \exp\!\Big(-\int_0^\tau f(u)\, du\Big), \qquad \sigma(\tau)^2 \;:=\; \int_0^\tau g(u)^2\, s(u)^{-2}\, du, \qquad \tau \geq 0,$$

*which is also stated in* (2). *For every $\tau > 0$ and every frequency $\boldsymbol{\xi} \in \mathbb{R}^d$ the Fourier transform of the solution to*

$$\partial_\tau p_\tau(\boldsymbol{x}) \;=\; \tfrac{1}{2}\, g(\tau)^2\, \Delta p_\tau(\boldsymbol{x}) \;-\; \nabla \cdot \big[ f(\tau)\, \boldsymbol{x}\, p_\tau(\boldsymbol{x}) \big], \quad p\big|_{\tau=0} \;=\; p_0,$$

*satisfies the identity*

$$\widehat{p_\tau}(\boldsymbol{\xi}) \;=\; \widehat{p_0}\big(\boldsymbol{\Xi}(0)\big)\, \exp\!\Big( -\tfrac{1}{2} \int_0^\tau g(\gamma)^2\, \big\| \boldsymbol{\Xi}(\gamma) \big\|^2\, \mathrm{d}\gamma \Big).$$

*Consequently,*

$$\big| \widehat{p_\tau}(\boldsymbol{\xi}) \big| \;\leq\; \big| \widehat{p_0}\big(\boldsymbol{\Xi}(0)\big) \big|\, \exp\!\Big( -\tfrac{1}{2} \int_0^\tau g(\gamma)^2\, \big\| \boldsymbol{\Xi}(\gamma) \big\|^2\, \mathrm{d}\gamma \Big).$$

*Proof.* Step 1: Fourier transform of the Fokker–Planck operator. Because $p_\tau \in L^1 \cap L^2$ and enjoys Gaussian upper bounds (see assumption (A3)), we apply the spatial Fourier transform and integrate by parts. Writing $\widehat{p_\tau}(\boldsymbol{\xi}) = \mathcal{F}_{\boldsymbol{x}}[p_\tau](\boldsymbol{\xi})$, we use the standard identities:

$$\mathcal{F}[\Delta p_\tau](\boldsymbol{\xi}) \;=\; -\|\boldsymbol{\xi}\|^2\, \widehat{p_\tau}(\boldsymbol{\xi}), \qquad \mathcal{F}\Big[ \nabla \cdot (\boldsymbol{x}\, p_\tau) \Big](\boldsymbol{\xi}) \;=\; -\big\langle \boldsymbol{\xi},\, \nabla_{\boldsymbol{\xi}}\, \widehat{p_\tau}(\boldsymbol{\xi}) \big\rangle.$$

Transforming the Fokker–Planck equation therefore gives the first-order PDE

$$\partial_\tau \widehat{p_\tau}(\boldsymbol{\xi}) \;=\; -\tfrac{1}{2}\, g(\tau)^2\, \|\boldsymbol{\xi}\|^2\, \widehat{p_\tau}(\boldsymbol{\xi}) \;+\; f(\tau) \big\langle \boldsymbol{\xi},\, \nabla_{\boldsymbol{\xi}}\, \widehat{p_\tau}(\boldsymbol{\xi}) \big\rangle,$$

to be solved in the $(\tau, \boldsymbol{\xi})$ variables. Or, equivalently

$$\partial_\tau \widehat{p_\tau}(\boldsymbol{\xi}) \;-\; f(\tau) \big\langle \boldsymbol{\xi},\, \nabla_{\boldsymbol{\xi}} \widehat{p_\tau}(\boldsymbol{\xi}) \big\rangle \;=\; -\tfrac{1}{2}\, g(\tau)^2\, \|\boldsymbol{\xi}\|^2\, \widehat{p_\tau}(\boldsymbol{\xi}).$$

Step 2: Solve the equation via characteristic curves in $\boldsymbol{\xi}$-space. Fix $\tau > 0$ and $\boldsymbol{\xi} \in \mathbb{R}^d$. For $\gamma \in [0, \tau]$ define the backward characteristic $\boldsymbol{\Xi}(\gamma)$ by

$$\frac{d\boldsymbol{\Xi}}{d\gamma}(\gamma) = -f(\gamma)\,\boldsymbol{\Xi}(\gamma), \qquad \boldsymbol{\Xi}\big|_{\gamma=\tau} = \boldsymbol{\xi}.$$

Since $f \in L^1_{\text{loc}}([0, \infty))$ by Assumption (A2), the ODE is well posed and has the explicit solution

$$\boldsymbol{\Xi}(\gamma) = \exp\Big(-\int_\gamma^\tau f(u)\,\mathrm{d}u\Big)\boldsymbol{\xi} = \frac{s(\tau)}{s(\gamma)}\boldsymbol{\xi}, \qquad 0 \le \gamma \le \tau,$$

where $s(\cdot)$ is the scaling factor defined in (2). Along this curve in $\boldsymbol{\xi}$-space, using the chain rule and the PDE obtained in Step 1 we compute:

$$\frac{d}{d\gamma}\,\widehat{p_\gamma}(\boldsymbol{\Xi}(\gamma)) = -\tfrac{1}{2}\,g(\gamma)^2\,\big\|\boldsymbol{\Xi}(\gamma)\big\|^2\,\widehat{p_\gamma}(\boldsymbol{\Xi}(\gamma)).$$

This linear ODE integrates to

$$\widehat{p_\gamma}(\boldsymbol{\Xi}(\gamma)) = \widehat{p_0}(\boldsymbol{\Xi}(0))\,\exp\Big(-\tfrac{1}{2}\int_0^\gamma g(u)^2\,\big\|\boldsymbol{\Xi}(u)\big\|^2\,\mathrm{d}u\Big), \qquad 0 \le \gamma \le \tau.$$

Setting $\gamma = \tau$ in the equation obtained above, we have

$$\widehat{p_\tau}(\boldsymbol{\xi}) = \widehat{p_0}(\boldsymbol{\Xi}(0))\,\exp\Big(-\tfrac{1}{2}\int_0^\tau g(u)^2\,\big\|\boldsymbol{\Xi}(u)\big\|^2\,\mathrm{d}u\Big).$$

Recall that $\boldsymbol{\Xi}(u) = \frac{s(\tau)}{s(u)}\boldsymbol{\xi}$ where $s(\tau) = \exp\big(-\int_0^\tau f(u)\,du\big)$. Since $s(0) = 1$, we have

$$\boldsymbol{\Xi}(0) = s(\tau)\,\boldsymbol{\xi} \quad \text{and} \quad \widehat{p_0}(\boldsymbol{\Xi}(0)) = \widehat{p_0}\big(s(\tau)\,\boldsymbol{\xi}\big).$$

For the integral in the exponent, we use $\|\boldsymbol{\Xi}(u)\|^2 = \big(\frac{s(\tau)}{s(u)}\big)^2 \|\boldsymbol{\xi}\|^2$ to obtain

$$\int_0^\tau g(u)^2\,\|\boldsymbol{\Xi}(u)\|^2\,du = \|\boldsymbol{\xi}\|^2 s(\tau)^2 \int_0^\tau \frac{g(u)^2}{s(u)^2}\,du.$$

Defining $\sigma(\tau)^2 := \int_0^\tau \frac{g(u)^2}{s(u)^2}\,du$, this simplifies to

$$\int_0^\tau g(u)^2\,\|\boldsymbol{\Xi}(u)\|^2\,du = \|\boldsymbol{\xi}\|^2 s(\tau)^2 \sigma(\tau)^2.$$

Substituting these results into our expression for $\widehat{p_\tau}(\boldsymbol{\xi})$ yields

$$\widehat{p_\tau}(\boldsymbol{\xi}) = \widehat{p_0}(s(\tau)\boldsymbol{\xi})\,\exp\Big(-\frac{1}{2}\|\boldsymbol{\xi}\|^2 s(\tau)^2 \sigma(\tau)^2\Big),$$

which matches the identity in Theorem A.1 $\qquad\qquad\square$

**Takeaway.**  The final expression

$$\widehat{p_\tau}(\boldsymbol{\xi}) = \widehat{p_0}\big(s(\tau)\,\boldsymbol{\xi}\big)\,\exp\Big(-\tfrac{1}{2}\,\|\boldsymbol{\xi}\|^2\,s(\tau)^2\,\sigma^2(\tau)\Big)$$

shows that for each fixed $\tau > 0$, the Fourier transform of $p_\tau$ inherits a *Gaussian-type decay factor* in $\|\boldsymbol{\xi}\|^2$. Provided $g(\cdot)$ does not vanish identically and $s(\cdot)$ remains finite and nonzero, the product $s(\tau)^2 \sigma^2(\tau)$ is strictly positive, thus giving an exponential decay in $\|\boldsymbol{\xi}\|$ for each $t > 0$. In other words, *no matter how $f(\cdot)$ and $g(\cdot)$ vary over time*, the factor $\exp\big(-\tfrac{1}{2}\|\boldsymbol{\xi}\|^2\,s(\tau)^2\,\sigma(\tau)^2\big)$ suppresses large-frequency modes of $p_\tau$ in a Gaussian manner.

Moreover, we can view this suppression in *two complementary ways*: **(a)** *Decay in frequency for fixed $\tau$.* For any fixed $\tau > 0$, as $\|\boldsymbol{\xi}\| \to \infty$, we have the factor $\exp\big(-\tfrac{1}{2}\|\boldsymbol{\xi}\|^2\,s(\tau)^2\,\sigma(\tau)^2\big)$ forcing rapid (Gaussian) damping of high-frequency modes. This is the source of *instant smoothing*: one can show $p_\tau \in H^\kappa(\mathbb{R}^d)$ for all $\kappa \ge 0$. **(b)** *Decay in time for fixed $\boldsymbol{\xi} \ne 0$.* For any nonzero frequency $\boldsymbol{\xi}$, the same exponential factor shrinks $\widehat{p_\tau}(\boldsymbol{\xi})$ to zero in $\tau$, typically at an exponential rate, provided $s(\tau)^2 \sigma^2(\tau)$ grows in $\tau$. This illustrates how each individual mode is damped as time evolves.

These two types of decay reflect the main *smoothing* and *mode-attenuating* nature of forward diffusion processes. **(a)** *First*, $p_\tau$ becomes *instantly smooth* for any $\tau > 0$: indeed, $p_\tau \in H^\kappa(\mathbb{R}^d)$ for all $\kappa \ge 0$, implying $p_\tau$ is $C^\infty$ in the spatial variables. **(b)** *Second*, from an inverse perspective (e.g. generative modeling), the *loss* of high-frequency content means that recovering $p_0$ from $p_\tau$ becomes ill-conditioned as $\tau$ grows. Thus this exponential decay factor captures both the *smoothing* and the *mode-attenuating* nature of forward diffusion processes.

## A.2 Instant smoothing in Sobolev and Wavelet spaces

In the previous subsection, we established that $\widehat{p_\tau}(\boldsymbol{\xi})$ decays exponentially at large wavenumbers $\|\boldsymbol{\xi}\|$ (see Theorem A.1). Here, we show how this exponential decay in the Fourier domain leads directly to the conclusion that $p_\tau$ belongs to *all* Sobolev spaces $H^\kappa(\mathbb{R}^d)$ for each $\tau > 0$. Then, by leveraging the known equivalence between Sobolev norms and wavelet-coefficient decay, we confirm that $p_\tau$ is likewise instantly smooth with respect to any transform that localizes high-frequency (or fine-scale) components, such as wavelets.

**Corollary A.2** (Instant smoothing in Sobolev and Wavelet spaces)**.** *Let $p_\tau$ solve the linear-drift Fokker–Planck equation* (5) *and assume that $p_\tau(\cdot)$ remains in $L^2(\mathbb{R}^d)$ for all $\tau \geq 0$ (see Assumption* (A3)*). Then for each fixed $\tau > 0$ and for* every *real $\kappa \geq 0$, we have*

$$p_\tau \ \in \ H^\kappa(\mathbb{R}^d).$$

*Equivalently, in a wavelet basis $\{\psi_{j,k}\}$, if $b_{j,k}(p_\tau)$ denotes the wavelet coefficients of $p_\tau$, then*

$$\sum_{j \in \mathbb{Z}} \sum_{k \in \mathbb{Z}^d} 2^{2j\kappa} \left| b_{j,k}(p_\tau) \right|^2 \ < \ \infty \quad \text{for each } \kappa \geq 0.$$

*Hence $p_\tau$ is in* all *Sobolev space $H^\kappa(\mathbb{R}^d)$ for $\tau > 0$, and is therefore infinitely differentiable.*

*Proof of Corollary A.2.* Step 1: Exponential decay in the Fourier domain. By Theorem A.1, there exist constants $C_\tau > 0$, $\alpha_\tau > 0$, and $R_0 > 0$, depending on $\tau$, such that

$$\left| \widehat{p_\tau}(\boldsymbol{\xi}) \right| \ \leq \ C_\tau \, \exp\!\left( - \alpha_\tau \|\boldsymbol{\xi}\|^2 \right) \quad \text{for all } \|\boldsymbol{\xi}\| \ \geq \ R_0.$$

(If $f(\cdot) \equiv 0$, this is immediately seen from the heat kernel's Gaussian decay. If $f(\cdot) \neq 0$, one uses the scaled-argument form of the solution in Theorem A.1.)

Step 2: Sobolev-norm boundedness. Recall that $p_\tau \in H^\kappa(\mathbb{R}^d)$ for all $\kappa > 0$ if and only if

$$\int_{\mathbb{R}^d} \left( 1 + \|\boldsymbol{\xi}\|^2 \right)^\kappa \left| \widehat{p_\tau}(\boldsymbol{\xi}) \right|^2 d\boldsymbol{\xi} \ < \ \infty. \tag{$*$}$$

(See, e.g., Grafakos [2014], Evans [1998] for this classical characterization.) We split the integral in $(*)$ into two regions:

$$\{\|\boldsymbol{\xi}\| \leq R_0\} \quad \text{and} \quad \{\|\boldsymbol{\xi}\| > R_0\}.$$

Since $\widehat{p_\tau} \in L^2(\mathbb{R}^d)$ (by Assumption (A3), $p_\tau \in L^2(\mathbb{R}^d)$ and Plancherel's theorem), the portion over $\{\|\boldsymbol{\xi}\| \leq R_0\}$ is trivially finite. On $\{\|\boldsymbol{\xi}\| > R_0\}$, we use the exponential bound:

$$(1 + \|\boldsymbol{\xi}\|^2)^\kappa \left| \widehat{p_\tau}(\boldsymbol{\xi}) \right|^2 \ \leq \ (1 + \|\boldsymbol{\xi}\|^2)^\kappa \, C_\tau^2 \, \exp\!\left( -2 \, \alpha_\tau \|\boldsymbol{\xi}\|^2 \right).$$

As $\|\boldsymbol{\xi}\| \to \infty$, the polynomial factor $(1 + \|\boldsymbol{\xi}\|^2)^\kappa$ is dominated by the superpolynomial decay of $\exp\!\left( -2 \, \alpha_\tau \|\boldsymbol{\xi}\|^2 \right)$. Hence the integrand is integrable for all $\kappa \geq 0$. Thus the integral in $(*)$ is finite, yielding $p_\tau \in H^\kappa(\mathbb{R}^d)$. Since $\kappa$ is arbitrary, $p_\tau \in H^\kappa(\mathbb{R}^d)$ for *all* $\kappa \geq 0$.

Step 3: Equivalence with wavelet decay. Next, we recall (e.g. from Meyer [1993]) that a function's inclusion in $H^\kappa(\mathbb{R}^d)$ is *equivalent* to having sufficiently small wavelet coefficients at large scales:

$$\|p_\tau\|_{H^\kappa(\mathbb{R}^d)}^2 \ \sim \ \sum_{j \in \mathbb{Z}} \sum_{k \in \mathbb{Z}^d} 2^{2j\kappa} \left| b_{j,k}(p_\tau) \right|^2,$$

where $b_{j,k}(p_\tau)$ denotes the wavelet coefficient of $f$ at scale $j$ and position $k$. Since exponential decay of $\widehat{p_\tau}(\boldsymbol{\xi})$ at large $\|\boldsymbol{\xi}\|$ implies fast convergence of $p_\tau$'s expansions in localized bases (such as wavelets), one sees directly that $p_\tau$'s wavelet coefficients decay sufficiently rapidly for it to lie in all $H^\kappa$ for $\kappa > 0$. Equivalently, from the purely wavelet viewpoint, each increase in $\kappa$ imposes an extra factor of $2^{2j\kappa}$ on the sum, yet exponential decay in frequency/scale ensures convergence for every $\kappa$.

In conclusion, $p_\tau$ lies in all $H^\kappa(\mathbb{R}^d)$ and thus is $C^\infty(\mathbb{R}^d)$ for each $\tau > 0$.

$\square$

*Remark* A.3 (Implications for generative models). From the perspective of generative modeling, Theorem A.1 and Corollary A.2 formalize the idea that "noise injection" destroys high-frequency information about the initial distribution $p_0$. The exponential shrinkage of $\widehat{p_\tau}(\boldsymbol{\xi})$ at large $\|\boldsymbol{\xi}\|$ makes recovery of $p_0$ from $p_\tau$ ill-conditioned as $\tau$ increases. Hence effective methods may require learned priors or neural architectures that reconstruct the lost high-frequency details in a stable way. From a practical standpoint, the generative *reverse* process must reconstruct precisely those exponentially damped (high) frequencies, which explains in part why carefully designed neural network architectures and training procedures are needed.

*Remark* A.4 (Conditions for smoothing properties). Consider the forward SDE defined in equation (1). **(a)** $C^\infty$-*smoothness.* When the diffusion coefficient is *strictly non-degenerate*, i.e., $g(\tau) > 0$ for all $\tau > 0$, Theorem A.1 establishes that $\widehat{p_\tau}(\boldsymbol{\xi}) = \mathcal{O}(e^{-c_\tau \|\boldsymbol{\xi}\|^2})$. Consequently, $p_\tau \in H^\kappa(\mathbb{R}^d)$ for every $\kappa \geq 0$ and exhibits $C^\infty$-smoothness with respect to $\boldsymbol{x}$. The strict positivity condition on $g(\tau)$ is necessary; in cases where $g \equiv 0$ (purely deterministic flow) or where $g$ is rank-deficient and Hörmander's bracket condition is not satisfied, $p_\tau$ may reduce to a Dirac distribution without any smoothing effect. **(b)** *Real analyticity.* If, furthermore, the coefficients satisfy *uniform parabolicity* on each compact time interval,

$$0 < \inf_{0 \leq \gamma \leq \sigma} g(\gamma) \leq \sup_{0 \leq \gamma \leq \sigma} g(\gamma) < \infty, \quad \sup_{0 \leq \gamma \leq \sigma} |f(\gamma)| < \infty \quad (\forall\, \sigma > 0),$$

then we can strengthen the regularity conclusion to real-analyticity of $p_\tau$ in the spatial variables for all $\tau > 0$. **(c)** *Practical implication.* Many score-based diffusion models (e.g., EDMKarras et al. [2022]) specifically choose a full-rank Brownian motion with $g$ both bounded and bounded away from zero. Consequently, both the $C^\infty$-smoothness property and the analytic regularity are automatically satisfied in practical implementations.

## B   Algorithm

In the following, we collect the implementation details that turn the approach in Secs.4–4.3 into a step-by-step algorithm. The algorithm 1 details the two-stage optimization that equips the inverse operator $\phi$ with the boundary condition and the semigroup constraint that connects all intermediate noise levels. The algorithm 2 illustrates how, once trained, the same operator can be used to generate (or complete) time series in a single jump from the highest noise level $\sigma_{\max}$ to the data manifold at $\sigma = 0$.

## C   Experiments

**Datasets.**   We evaluate our model on two main tasks: forecasting and imputation. For forecasting, we utilize five benchmark time series datasets: Electricity Transformer Temperature (ETT) Zhou et al. [2021], which contains electricity transformer measurements at hourly (ETTh) and 15-minute (ETTm) intervals; Weather Wu et al. [2021], comprising 21 meteorological parameters collected at 10-minute intervals; Solar-Energy Lai et al. [2017], which records power generation data from multiple solar plants at 10-minute intervals throughout 2006; Electricity Wu et al. [2021], tracking consumption patterns across 321 clients; and Traffic Wu et al. [2021], which monitors hourly road occupancy rates through 862 sensors in San Francisco from 2015 to 2016. For the imputation task, we use a subset of these forecasting datasets along with Energy Candanedo [2017], which captures energy consumption at 10-minute intervals over approximately 4.5 months, collected via a ZigBee wireless sensor network in Belgium.

**Experimental setup.**   Our experimental configurations follow the protocols established in Wu et al. [2023], including identical data processing and splitting methods. Details for each dataset are provided in Table 4. Aligned with fair comparison settings outlined in Liu et al. [2024], Li et al. [2025], we fix the lookback window length to 96 time steps for most datasets and baselines, with the prediction horizon set to 192 time steps. For baselines that are reported under similar settings and have published results for 192-step prediction, such as NsDiffYe et al. [2025], we adopt their default configurations, regardless of whether they utilize longer lookback windows (which potentially provide more historical context for forecasting or conditioned generation). For imputation tasks, we fix the window length to 48 time steps, following the protocol in Yuan and Qiao [2024]. For probabilistic

---

**Algorithm 1** Two-stage training of the single–step inverse operator $\phi$

---

**Input:** dataset $\mathcal{D}$ of clean sequences $\boldsymbol{x}_0$; mini-batch size $N$; log–normal noise prior $p_{ln}(\sigma)$; weighting function $\omega(\sigma)$ (Eq. 4); learning rates $\eta_1, \eta_2$; EMA rate $\beta_{\text{ema}}$; adaptive linear grid scheduler $N(\cdot) = \text{identity}(\cdot)$; noise scheduler in second stage $\sigma_n | N(k) = (\sigma_{\max}^{1/\upsilon} + \frac{n}{N(k)-1}(\sigma_{\min}^{1/\upsilon} - \sigma_{\max}^{1/\upsilon}))^\upsilon$ with $\sigma_{\max} = 80$, $\sigma_{\min} = 0.002$, and $\upsilon = 7$

1   **Initialize:** random parameters $\theta$ (shared by $H_\theta$ in Eq. (3) and $\phi$ in Eq. (8)); $\bar{\theta} \leftarrow \theta$

    **Stage 1: boundary denoising (Eq. (4))**

2  **while** not converged **do**

3     Sample $\{\boldsymbol{x}_0^{(i)}\}_{i=1}^N \sim \mathcal{D}$

4     Sample $\{\sigma^{(i)}\}_{i=1}^N \sim p_{ln}(\sigma)$ and $\{\boldsymbol{\epsilon}^{(i)}\}_{i=1}^N \sim \mathcal{N}(\mathbf{0}, \mathbf{I})$

5     **for** $i = 1$ **to** $N$ **do**

6         $\boldsymbol{x}_\sigma^{(i)} \leftarrow \boldsymbol{x}_0^{(i)} + \sigma^{(i)} \boldsymbol{\epsilon}^{(i)}$

7         $\hat{\boldsymbol{x}}_0^{(i)} \leftarrow H_\theta(\boldsymbol{x}_\sigma^{(i)}, \sigma^{(i)})$

8     **end for**

9     $\mathcal{L}_{\text{denoise}} \leftarrow \dfrac{1}{N} \sum_{i=1}^{N} \omega(\sigma^{(i)}) \big\| \hat{\boldsymbol{x}}_0^{(i)} - \boldsymbol{x}_0^{(i)} \big\|_2^2$

10    $\theta \leftarrow \theta - \eta_1 \nabla_\theta \mathcal{L}_{\text{denoise}}$

11 **end while**

    **Stage 2: semigroup consistency (Eq. (9))**

12 **while** not converged **do**

13    Increment global step $k$

14    Sample $\{\boldsymbol{x}_0^{(i)}\}_{i=1}^N \sim \mathcal{D}$

15    Choose $n \sim \text{Unif}\{1, \ldots, N(k)\}$; set $\tau \leftarrow \sigma_n, \gamma \leftarrow \sigma_{n-1}, \rho \leftarrow 0$

16    Sample $\{\boldsymbol{\epsilon}^{(i)}\}_{i=1}^N \sim \mathcal{N}(\mathbf{0}, \mathbf{I})$

17    **for** $i = 1$ **to** $N$ **do**

18       $\boldsymbol{x}_\tau^{(i)} \leftarrow \boldsymbol{x}_0^{(i)} + \tau \boldsymbol{\epsilon}^{(i)}$

19       $\boldsymbol{x}_{\text{direct}}^{(i)} \leftarrow \phi_\theta(\rho, \tau, \boldsymbol{x}_\tau^{(i)})$

20       $\boldsymbol{x}_{\text{comp}}^{(i)} \leftarrow \phi_\theta\big(\rho, \gamma, \phi_{\bar{\theta}}(\gamma, \tau, \boldsymbol{x}_\tau^{(i)})\big)$

21    **end for**

22    $\mathcal{L}_{\text{cmp}} \leftarrow \dfrac{1}{N} \sum_{i=1}^{N} \big\| \boldsymbol{x}_{\text{direct}}^{(i)} - \boldsymbol{x}_{\text{comp}}^{(i)} \big\|_2^2$

23    $\theta \leftarrow \theta - \eta_2 \nabla_\theta \mathcal{L}_{\text{cmp}}$

24    $\bar{\theta} \leftarrow \beta_{\text{ema}} \bar{\theta} + (1 - \beta_{\text{ema}}) \theta$

25 **end while**

**Return:** $\bar{\theta}$                  $\triangleright$ final EMA parameters

---

---

**Algorithm 2** Single–step sampling with inverse operator $\phi$

---

**Input:** number of samples $\tilde{N}$; trained EMA parameters $\bar{\theta}$; maximum noise level $\sigma_{\max}$; observations $\boldsymbol{x}_{\text{obs}}$, mask $\mathbf{M}$

**Return:** completed sequences $\{\widehat{\boldsymbol{x}}_0^{(i)}\}_{i=1}^{\tilde{N}}$

1  **for** $i = 1$ **to** $\tilde{N}$ **do**                 $\triangleright$ independent draws

2     draw $\boldsymbol{\epsilon}^{(i)} \sim \mathcal{N}(\mathbf{0}, \mathbf{I})$

3     $\boldsymbol{x}_\sigma^{(i)} \leftarrow \sigma_{\max} \boldsymbol{\epsilon}^{(i)}$             $\triangleright$ initialise at highest noise

4     **if** observations are given **then**         $\triangleright$ completion case

5         $\boldsymbol{x}_\sigma^{(i)} \leftarrow (1 - \mathbf{M}) \odot \boldsymbol{x}_\sigma^{(i)} + \mathbf{M} \odot \boldsymbol{x}_{\text{obs}}$

6     **end if**

7     $\widehat{\boldsymbol{x}}_0^{(i)} \leftarrow \phi_{\bar{\theta}}(0, \sigma_{\max}, \boldsymbol{x}_\sigma^{(i)})$         $\triangleright$ single jump $\sigma_{\max} \to 0$

8  **end for**

    **return** $\{\widehat{\boldsymbol{x}}_0^{(i)}\}_{i=1}^{\tilde{N}}$

---

completion tasks (including both forecasting and imputation), we sample 100 trajectories per instance and use the median trajectory as our point estimator.

Table 4: Dataset characteristics for forecasting and imputation tasks. *Dim.* denotes the number of variates (channels); *Size* specifies the sample counts in the training, validation, and test splits; *Freq.* indicates the data sampling interval.

| Task | Dataset | Dim. | Pred. length/Missing ratio | Size | Freq. | Domain |
|------|---------|------|---------------------------|------|-------|--------|
| Forecasting | ETTh1 | 7 | 192 | (8545, 2881, 2881) | 1 hr | Temperature |
| | ETTm1 | 7 | 192 | (34465, 11521, 11521) | 15 min | Temperature |
| | ETTm2 | 7 | 192 | (34465, 11521, 11521) | 15 min | Temperature |
| | Weather | 21 | 192 | (36792, 5271, 10540) | 10 min | Weather |
| | Electricity | 321 | 192 | (18317, 2633, 5261) | 1 hr | Electricity |
| | Solar-Energy | 137 | 192 | (36601, 5161, 10417) | 10 min | Electricity |
| | Traffic | 862 | 192 | (12185, 1757, 3509) | 1 hr | Transportation |
| Imputation | ETTh1 | 7 | {10,25,50,75,90}% | (8545, 2881, 2881) | 1 hr | Temperature |
| | ETTm1 | 7 | {10,25,50,75,90}% | (34465, 11521, 11521) | 15 min | Temperature |
| | Weather | 21 | {10,25,50,75,90}% | (36792, 5271, 10540) | 10 min | Weather |
| | Energy | 28 | {10,25,50,75,90}% | (13797, 1972, 3942) | 10 min | Sensor network |

**Baselines.** For the time series forecasting task (conditioned generation with masks identifying the lookback and prediction horizon), we conducted comprehensive comparisons with both point forecasting models and probabilistic models. The point forecasting models include NSformer Liu et al. [2022], TimesNet Wu et al. [2023], DLinear Zeng et al. [2023], PatchTST Nie et al. [2023], SparseVQ Zhao et al. [2024], and iTransformer Liu et al. [2024]. The probabilistic models include TimeGrad Rasul et al. [2021], CSDI Tashiro et al. [2021], TimeDiff Shen and Kwok [2023], DiffusionTS Yuan and Qiao [2024], TMDM Li et al. [2024], D3U Li et al. [2025], and NsDiff Ye et al. [2025]. For models that explicitly addressed this forecasting task under mostly identical settings, specifically D3U Li et al. [2025] and NsDiff Ye et al. [2025], we directly referenced their published results to ensure accuracy and consistency in our comparisons. For baselines that did not originally report results under our exact experimental conditions, we either reproduced their outcomes using official code repositories where available, or referenced reproduction results reported by D3U and NsDiff to maintain comparability and fairness across all evaluations.

**Implementation details.** All experiments were conducted using PyTorch Paszke et al. [2019] on a single NVIDIA A100 40GB GPU. We trained our model using a two-stage approach with the Adam optimizer Kingma and Ba [2015]. In the first stage, we trained the model using the EDM-type loss Karras et al. [2022] defined in Eq. 4. The second stage focused on optimizing the composition property loss, as shown in Algorithm 1. Summary of the experimental configurations for all datasets is listed in Table 5.

Table 5: Hyperparameter settings used for training and evaluating the diffusion models.

| Dataset | Epochs | Batch Size | Learning Rate | Layers | Wavelet Levels | Wavelet Type | Sampled Trajectories | Aggregation |
|---------|--------|-----------|---------------|--------|----------------|--------------|---------------------|-------------|
| ETTm1 | 50 | 32 | 0.001 | 4 | 3 | db1 | 100 | median |
| ETTm2 | 50 | 32 | 0.001 | 4 | 1 | db1 | 100 | median |
| Solar-Energy | 50 | 8 | 0.001 | 4 | 3 | db8 | 100 | median |
| Electricity | 50 | 8 | 0.001 | 4 | 3 | db1 | 100 | median |
| Weather | 50 | 32 | 0.001 | 4 | 3 | db1 | 100 | median |
| Traffic | 50 | 16 | 0.001 | 1 | 1 | db1 | 100 | median |
| Energy | 50 | 32 | 0.001 | 4 | 3 | db1 | 100 | median |

**Probabilistic metrics.** For evaluating probabilistic predictions, we employ the Continuous Ranked Probability Score (CRPS) [Matheson and Winkler, 1976, Tashiro et al., 2021], which measures compatibility between an estimated distribution and observations. CRPS is defined as:

$$\text{CRPS}(\mathbb{P}^{-1}, \boldsymbol{x}) = \int_0^1 2\Lambda_u(\mathbb{P}^{-1}(u), \boldsymbol{x})du \tag{10}$$

where $\Lambda_u(q, \boldsymbol{x}) = (u - \mathbb{1}_{\boldsymbol{x}<q})(\boldsymbol{x} - q)$ is the quantile loss function, $\mathbb{P}^{-1}$ is the inverse CDF of the predicted distribution, and $\boldsymbol{x}$ is the observed value. In practice, we approximate this integral using discretized quantile levels with 0.05 intervals and generate 100 samples to estimate the distribution. For multivariate time series, we also evaluate CRPS-sum, which assesses the joint predictive performance

by calculating CRPS for the distribution of the sum across all channels:

$$\text{CRPS-sum} = \frac{\sum_t \text{CRPS}(\mathbb{P}^{-1}, \sum_c \boldsymbol{x}_{c,t})}{\sum_{c,t} |\boldsymbol{x}_{c,t}|} \tag{11}$$

where $c$ indexes channels and $t$ indexes position in timesteps. Both metrics are normalized by the sum of absolute observed values to ensure fair comparison across datasets, with lower values indicating better model performance.

## C.1 Additional experiments in probabilistic forecasting

**Probabilistic metrics.** While our main evaluation (Table 1) employed deterministic metrics (MSE/MAE), we further assess probabilistic forecasting performance. Table 2 presents both CRPS and CRPS$_{\text{sum}}$ metrics, evaluating distributional accuracy at individual channel and aggregate levels, respectively.

Analysis of Table 2 reveals that SSOL achieves state-of-the-art CRPS$_{\text{sum}}$ performance on four of six datasets, demonstrating superior accuracy in aggregated channel forecasting. This advantage stems directly from our channel permutation invariance design. While D3U excels in per-channel CRPS, SSOL maintains competitive performance, ranking first on ETTm2 and second on several datasets. This performance pattern illustrates the trade-off between optimizing for individual channels versus aggregated behavior, confirming the effectiveness of our approach for multi-channel forecasting tasks.

**Statistical robustness.** We report the statistical significance of our model's performance by computing the standard deviation across 5 different consecutive random seeds during the sampling stage, while keeping the trained model fixed. As shown in Table 6, the standard deviations are consistently small (on the order of $10^{-4}$ or $10^{-5}$), demonstrating that our results are robust to the stochasticity in the sampling process. This stability is expected since we use the median of 100 trajectories per instance as our point estimator, which provides statistical robustness.

Table 6: Mean$\pm$SD of each metric for prediction lengths 192 and 96 on ETTh1, computed over 5 random seeds during the sampling stage using the same trained model.

| Pred. Len | RMSE | MSE | MAE | CRPS | CRPS$_{\text{sum}}$ |
|---|---|---|---|---|---|
| 192 | $0.649 \pm 1.19 \times 10^{-4}$ | $0.421 \pm 1.54 \times 10^{-4}$ | $0.422 \pm 9.88 \times 10^{-5}$ | $0.405 \pm 6.76 \times 10^{-5}$ | $0.600 \pm 2.82 \times 10^{-4}$ |
| 96 | $0.612 \pm 1.43 \times 10^{-4}$ | $0.375 \pm 1.76 \times 10^{-4}$ | $0.396 \pm 1.17 \times 10^{-4}$ | $0.377 \pm 9.72 \times 10^{-5}$ | $0.564 \pm 2.05 \times 10^{-4}$ |

Table 7: Comparison of imputation performance with Diffusion-TS Yuan and Qiao [2024]. Metrics reported: MSE ($\times 10^{-3}$) and MAE ($\times 10^{-2}$). Diffusion-TS and SSOL (ours) results are based on 100 samples using the median as point estimator, while SSOL-S uses a single trajectory sample as estimator. ($\downarrow$)

| | | ETTh1 | | | | | Energy | | | | | Weather | | | | | ETTm1 | | | | |
|---|---|---|---|---|---|---|---|---|---|---|---|---|---|---|---|---|---|---|---|---|---|
| | | 10% | 25% | 50% | 75% | 90% | 10% | 25% | 50% | 75% | 90% | 10% | 25% | 50% | 75% | 90% | 10% | 25% | 50% | 75% | 90% |
| **Diffusion-TS** | MSE | 2.20 | 2.62 | 3.13 | 3.65 | 3.68 | 19.6 | 20.9 | 22.6 | 23.9 | 23.9 | 6.82 | 4.61 | 3.58 | 3.20 | 3.24 | 0.74 | 0.89 | 1.13 | 1.42 | 1.43 |
| | MAE | 3.11 | 3.35 | 3.71 | 4.06 | 4.07 | 10.0 | 10.2 | 10.4 | 10.8 | 10.8 | 1.64 | 1.37 | 1.25 | 1.22 | 1.23 | 2.71 | 2.99 | 3.36 | 3.76 | 3.79 |
| **SSOL** | MSE | **1.83** | **1.31** | **1.67** | **2.14** | **2.14** | **4.41** | **4.74** | **4.09** | **5.75** | **5.65** | **1.34** | **1.29** | **1.23** | **1.28** | **1.34** | **0.44** | **0.50** | **0.62** | **0.77** | **0.81** |
| | MAE | **2.56** | **2.40** | **2.68** | **3.05** | **3.03** | **2.81** | **2.90** | **2.52** | **3.07** | **2.85** | **0.90** | **0.87** | **0.78** | **0.79** | **0.83** | **1.45** | **1.49** | **1.61** | **1.78** | **1.89** |
| **SSOL-S** | MSE | 2.92 | 2.08 | 2.70 | 3.51 | 3.51 | 7.83 | 8.84 | 7.69 | 10.9 | 10.8 | 2.20 | 2.07 | 2.30 | 2.12 | 2.15 | 0.65 | 0.86 | 1.07 | 1.39 | 1.56 |
| | MAE | 3.31 | 3.08 | 3.51 | 3.99 | 3.97 | 3.43 | 3.74 | 3.15 | 3.99 | 3.72 | 1.21 | 1.16 | 1.14 | 1.09 | 1.14 | 1.79 | 1.99 | 2.17 | 2.44 | 2.64 |

**Runtime analysis.** We provide actual training and inference times measured on identical hardware to address efficiency claims in Table 8. We used a computationally efficient variant of our method with reduced model size and shallower residual layers, while preserving the core frequency-aware block and semigroup composition training methodology. Our two-stage training exhibits comparable per-iteration times to baseline methods, with Stage 1 (boundary denoising) and Stage 2 (semigroup constraints) adding minimal overhead compared to standard single-stage training. Although we do not achieve a pure 20× speedup in wall-clock time due to backbone architecture differences across methods, our approach shows strong efficiency gains during inference.

Table 8: Runtime analysis and computational efficiency comparison of SSOL, D3U, and NsDiff models on ETTh1 dataset for prediction lengths 336 and 720.

| Model | Pred. Len | Forecasting Performance | | | | Computational Efficiency | | | Resources |
| | | MSE $\downarrow$ | MAE $\downarrow$ | CRPS $\downarrow$ | $\text{CRPS}_{sum}$ $\downarrow$ | Train (s/iter) | Inference (min/batch) | NFEs | GPU Mem (MiB) |
|---|---|---|---|---|---|---|---|---|---|
| SSOL | 336 | 0.491 | 0.477 | 0.458 | 0.639 | 0.025 | 0.022 | 1 | 364 |
| | 720 | 0.539 | 0.535 | 0.508 | 0.672 | 0.032 | 0.044 | 1 | 646 |
| D3U | 336 | 0.512 | 0.478 | 0.351 | 0.922 | 0.025 | 0.078 | 20 | 582 |
| | 720 | 0.533 | 0.505 | 0.371 | 1.458 | 0.030 | 0.164 | 20 | 884 |
| NsDiff | 336 | 0.728 | 0.583 | 0.431 | 1.083 | 0.105 | 0.544 | 20 | 2560 |
| | 720 | 0.704 | 0.613 | 0.440 | 2.086 | 0.179 | 0.608 | 20 | 5380 |

The key findings are: **(a)** inference speedup of $3.5\times$ faster than D3U and $25\times$ faster than NsDiff, **(b)** memory efficiency with $35\%$ less GPU usage than D3U and $85\%$ less than NsDiff, and **(c)** performance parity, where, despite far fewer denoising steps, SSOL achieves comparable/better MSE/MAE. The efficiency gains come from our single-step design, while the additional training time for semigroup consistency is negligible compared to the inference savings.

**Longer horizons.** we conducted additional experiments on longer prediction horizons (336 and 720 steps) using the ETTh1 dataset, as shown in Table 8. The results demonstrate that our method maintains competitive performance at extended horizons: SSOL achieves comparable or better deterministic predictions with consistently better joint distribution modeling (CRPS-SUM improvements of 31% at 336 steps and 54% at 720 steps compared with D3U).

**Fourier versus wavelets.** We compared Fourier and wavelet transforms as the spectral basis in our frequency-aware block using ETTh1 with prediction length of 96. Wavelet transforms with db1 basis achieved better performance (MSE: 0.375, MAE: 0.396) compared to Fourier transforms (MSE: 0.391, MAE: 0.412). This difference aligns with expectations: Fourier analysis assumes global stationarity, while wavelets provide localized time-frequency analysis better suited for non-stationary time-series patterns.

## C.2 Additional experiments in imputation

**Experiment results.** We provide additional details on the time series imputation experiments shown in Figure 3. Our empirical evaluation demonstrates that the proposed SSOL method significantly outperforms existing time series imputation approaches, including the state-of-the-art Diffusion-TS model Yuan and Qiao [2024], across four evaluated datasets and missing ratios. As shown in Table 7, SSOL consistently achieves lower error metrics, with MSE reductions of 16-50% on ETTh1, 76-82% on Energy, 59-80% on Weather, and 40-44% on ETTm1 compared to Diffusion-TS. Both Diffusion-TS and our primary SSOL method use the median from the multiple trajectories as estimator, which is standard pipeline in time-series diffusion models. To further validate our approach, we also evaluate SSOL-S, a variant of our method that uses a single trajectory sample as the estimator rather than aggregating multiple samples. While SSOL-S primarily serves as a reference point, it remains competitive with or outperforms Diffusion-TS on several datasets, particularly on ETTm1 and Weather.

**Showcase of the imputation task.** The figure 5 demonstrates that our imputation method maintains high accuracy across varying missing data ratios (10% to 90%), the median imputation (red lines) almost accurately reconstructs long, continuous missing segments (regions with blue dots) by retaining the shape of the ground-truth curve (solid blue line) and staying within the 90% confidence interval (shaded red area) for most channels. This pattern suggests that our model effectively learns the temporal dependencies, such as periodic or seasonal trends, in each dataset and uses them to bridge prolonged gaps. The confidence bands expand around regions of sharp ground-truth fluctuations, indicating the model captures higher uncertainty in those transitions and reflects it in broader intervals.

The shaded areas represent credible intervals (5th-95th percentiles across 100 generated samples) that show the range of plausible imputed values given the observed data. Note that these differ from confidence intervals for the median estimator, which would be much narrower and indicate the precision of our point estimate rather than the inherent uncertainty in the imputation task. This pattern suggests that our model effectively learns the temporal dependencies, such as periodic or seasonal trends, in each dataset and uses them to bridge prolonged gaps. The confidence bands expand around regions of sharp ground-truth fluctuations, indicating the model captures higher uncertainty in those transitions and reflects it in broader intervals.

## D   Limitations and broader impact

**Limitations.**   Our theoretical results rely on standard regularity assumptions of linear drift and non-degenerate Gaussian diffusion (see Remark. A.4), which are satisfied by most modern diffusion frameworks Karras et al. [2022]. For multiplicative (state-dependent) noise models, which arise in variance-expanding or data-adaptive diffusion schedules, relaxing these assumptions and investigating the smoothing behavior of forward operators remains an open question. While our frequency-aware block effectively captures temporal patterns in regular time-series data, its extension to more complex frequency structures requires further investigation. Our empirical evaluation focuses on six publicly available multivariate time-series datasets with minute-to-hourly granularity. Adapting our single-step operator to irregularly sampled sequences, longer contexts (e.g., high-frequency financial data or audio signals), or entirely different modalities would require more non-trivial architectural modifications to properly capture domain-specific structures.

**Broader impact.**   Our single-step operator learning approach for conditioned time-series diffusion models reduces the number of function evaluations from a typical 20 Li et al. [2025], Ye et al. [2025] to just 1, yielding computational speedup during inference. This efficiency can translate to reduced energy consumption while maintaining comparable or superior performance across multiple benchmarks. The proposed method also naturally produces confidence intervals alongside point estimates, as shown in our showcase experiments (Fig. 5), allowing more informed decision-making in domains where trend identification is needed (e.g., energy forecasting, traffic monitoring). Since we focus on structured time-series data generation, potential negative impacts remain limited primarily to cases where historical data is biased or erroneous.

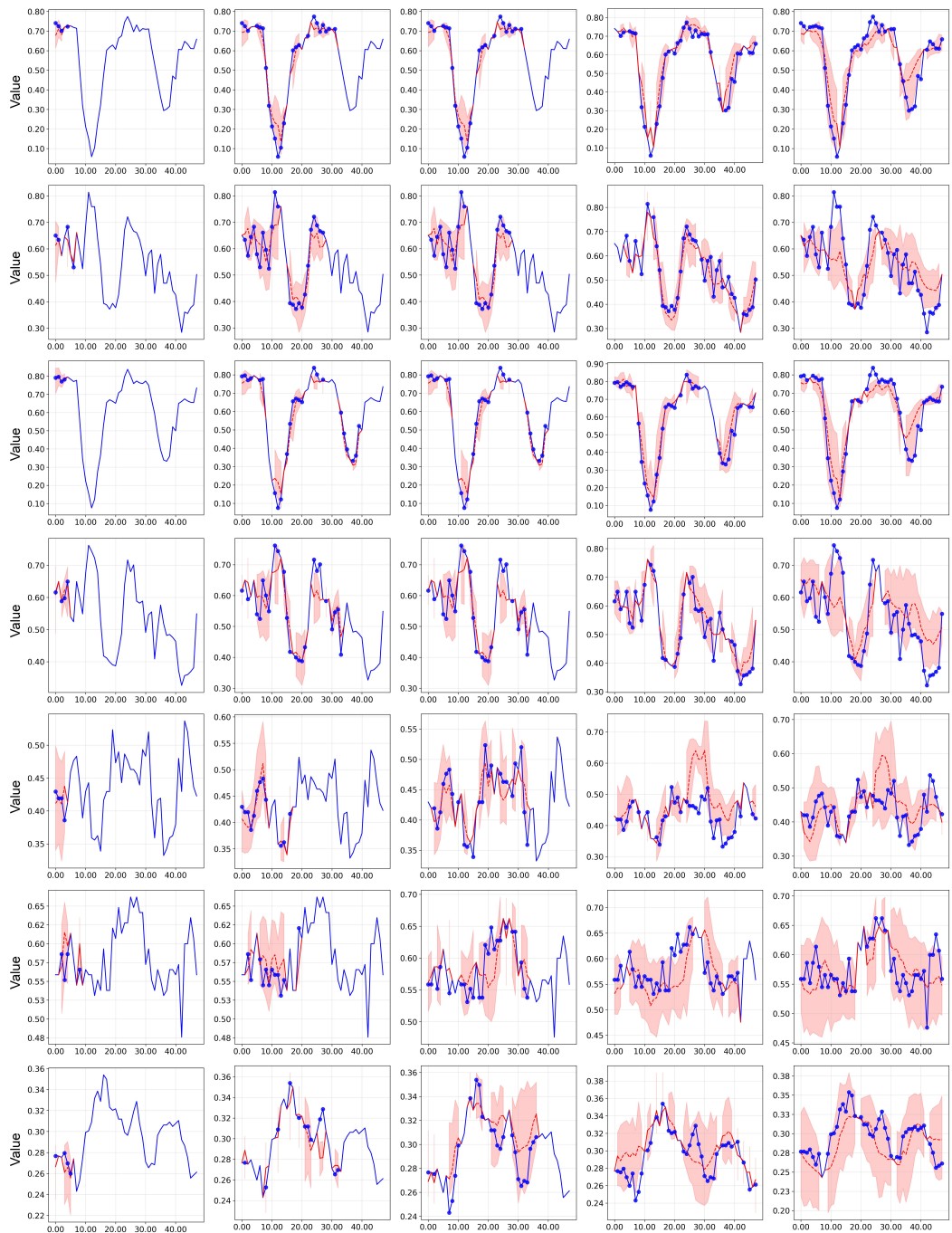

Figure 5: Imputation comparison for the ETTh1 dataset with a 48-step window using a time series sample of 7 features. The red shaded area represents the 90% confidence interval, the blue line denotes the ground truth, the dotted line indicates the median imputation, and the blue points mark the imputed ground truth values. Columns from left to right correspond to missing ratios of 10%, 25%, 50%, 75%, and 90%. Rows from top to bottom correspond to different channels.

