# OpenReview forum: "Single-Step Operator Learning for Conditioned Time-Series Diffusion Models"
_NeurIPS.cc/2025/Conference — NeurIPS 2025 poster_

### Official Review · Reviewer_bj9n · 2025-06-11

**Clarity:** 2
**Significance:** 4
**Originality:** 2
**Rating:** 4
**Confidence:** 4

**Summary:**

The paper proposes a model for time series. This is achieved by building on the observation that the noise-injection procedure in diffusion models removes/attenuates frequency components; therefore, the proposed model designs an operator over the frequency representation of the time series so as to ensure a correct treatment of the frequency components. Such operator, parametrised as a neural network, is implemented in a single step. The method is implemented over standard time series datasets and compared against a number of recent TS models.

**Questions:**

Major:
- It is not clear to this reviewer why the authors claim that (line 202) we need to **correct** the high-frequency damping introduced by DMs. That is the way DMs work, and it is not an "artefact", since intermediate (noisy) samples between source and target are not supposed to exactly follow the harmonic content of the data (but rather of the noisy data). I acknowledge that keeping track of the frequency components might be useful, but the call for avoiding their attenuation might be misleading
- Fig2: The figure is not clear:
    - What is the unnamed block in the bottom half of the figure? That block is disconnected from the rest of the diagram
    - there are symbols repeated in the figure (\hat{x}) - are those nodes the same?
    - The diagram shows a Fourier Transform (FT) operator, but the plot shows a power density (which is the square absolute value of the FT, not the FT)
- Some key elements in the paper are in the appendix only, e.g., the limitations and details about implementation
- The presentation of the method is done using the Fourier spectral representation, but in practice, the method is implemented using wavelets. Could the authors clarify and comment on this?
- One of the claimed contributions of the paper states: *significantly reducing computational overhead*, however, there is no experiment validating this claim. Note that the authors say that they refer to computational complexity in Appendix D, but this is largely uninformative.
- Fig 5 reveals that the error bars are rather large for the imputation experiment - some of them as large as the entire range of the signal. Also, the authors seem to use a fairly nonstandard presentation of their results, where blue dots are missing points (rather than observed points)
- Fig 3 (right): Why does the proposed method improve with more missing data (from 10% to 25%)
- The main experiments do not provide error bars, which makes it difficult to (statsitically) confirm the claimed superiority of the proposal
- The related work Section provides the right previous work; however, the authors do not place their own contribution in context
- Critically, the authors do not refer to the suitability of the analysis in the spectral domain. For instance, modelling the signal using the power spectrum (as suggested by Fig2) is only applicable for stationary signals. Furthermore, "coming back" from PSD space to time space requires signing the signal's phase, which is also unaddressed.




Minor:
- There seems to be a formatting issue with the references, the authors have probably typed \cite, but they use references as \citep
- line 344: Tabel -> Table
- Table II: only the year of the previous work is provided, and not the full reference.

**Ethical Concerns:**

["NO or VERY MINOR ethics concerns only"]

**Final Justification:**

I thank the authors for their responses to the points raised in my review. I am satisfied with the clarifications provided and thus increased my score.

**Limitations:**

See my questions above - also note that the limitations are only mentioned in the appendix

**Paper Formatting Concerns:**

problems with the way the references are cited (citep vs cite)

**Quality:**

2

**Strengths And Weaknesses:**

Though the underlying observation of the paper is a well-known feature of denoising DMs (even with blogs written on the topic, see the link below and the references therein), the proposal is interesting, but it can be explained and motivated more thoroughly. Also, the experimental validation of the method is very thorough and promising.

As far as I understand, a hypothesis of this work is that implementing the denoising operator in the frequency domain should preserve the structure of the time series in a way that is more accurate than if done in the time domain. Why is this the case? Shouldn't the frequency and temporal representation be equivalent? Since both representations are one-to-one, any operation defined over one space should have an equivalence in the other space.

I will refer to the weaknesses and strengths of the paper in more depth through questions in the next section.

blog: https://sander.ai/2024/09/02/spectral-autoregression.html

---

> ### Author Rebuttal · Authors · 2025-07-31
>
> We thank Reviewer bj9n for the thorough feedback and thoughtful questions. We have carefully addressed each point below.
>
> > Q1: Why do the authors claim the high-frequency damping from DMs needs correction?
> >
>
> You are absolutely correct that frequency attenuation during forward diffusion is the intended mechanism. We apologize for the sloppy language. To clarify: we are not trying to prevent or avoid the attenuation during forward diffusion. Rather, our goal is to efficiently reverse this attenuation in a single step during generation. The main point is that while individual noisy samples gain high-frequency noise (as expected), the probability density function undergoes systematic frequency-dependent smoothing (Theorem 3.2). In multi-step reverse diffusion, in general, this smoothing is gradually undone across many steps. Since we compress the entire reverse process into a single operator, our operator must learn to restore the full frequency spectrum in one transformation. Our FAB module helps the reverse operator perform this restoration by operating in frequency space where it can amplify the attenuated components based on the noise level. We will revise line 202 to clarify that we are addressing the reverse process, not claiming the forward attenuation is problematic. Thank you.
>
> > Q2: Fig.2 is not clear.
> >
> Thank you for the very detailed feedback! We will revise Figure 2 to address these issues:
>
> a) Unnamed block: This represents the semigroup consistency training (Stage 2 in Algorithm 1), where we enforce that composing two operations $\phi(\gamma,\tau,\cdot) \circ \phi(\rho,\gamma,\cdot)$ must equal the direct mapping $\phi(\rho,\tau,\cdot)$. We will add a clear label and connecting arrow to the main flow to avoid confusion.
>
> b) Symbol notation: The repeated $\hat{x}$ symbols means outputs at different stages, which we realize is also confusing. We will revise this: $x_\sigma$ for noisy inputs at noise level $\sigma$, and $\hat{x}_0$ for the final denoised output.
>
> c) FT vs Power Spectral Density: You are right: the visualization shows power spectral density as a cartoon illustration to just convey the concept of frequency-dependent processing. In practice, our FAB module operates on the full representation (both magnitude and phase for Fourier, or approximation and detail coefficients for wavelets). We will update the caption to clarify that this is a conceptual illustration, not the actual operation. Again, thanks for reading the figure carefully.
>
> > Q3: Some elements are in the appendix only.
> >
>
> This was primarily for space-saving reasons. If accepted, we will be allowed one additional page and this will allow us to absorb most of the implementation details and limitations into the main paper. Thanks.
>
> > Q4: Clarify the connection between the Fourier spectral theory and wavelet implementation.
> >
>
> Our technical analysis uses Fourier transforms because they provide a clearer/simpler description of frequency damping during diffusion (Theorem 3.2). However, as we show in Corollary 3.3, the same exponential damping and instant smoothing properties hold in any wavelet basis. For implementation, we chose wavelets because they are often preferred for time-series data: Fourier analysis assumes stationarity while wavelets can nicely capture time-localized patterns. But the important point is that our FAB module works with any orthogonal basis ${\chi_j}$. Both choices (Fourier/Wavelets) leverage the same underlying principle: diffusion attenuates high-frequency/fine-scale components that must be selectively restored during generation.
>
> > Q5: Claim of reducing computation overhead.
> >
>
> Thanks for encouraging us to emphasize this point. Our computational efficiency claim primarily comes from multi-step to single-step sampling. While some recent time-series diffusion models require 20 function evaluations during inference, SSOL achieves comparable results with a single denoising step. We performed additional runtime experiments and the key findings are as follows:
>
> (a) SSOL achieves 3.5x faster inference than D3U and 25x faster than NsDiff.
>
> (b) 35% less GPU memory than D3U, 85% less than NsDiff.
>
> (c) Despite far fewer denoising steps, SSOL achieves comparable/better MSE/MAE.
>
> The efficiency gains come from our single-step design. The additional training time for our second stage (semigroup consistency) is negligible compared to the inference savings. Additionally, as detailed in Table 2 in the manuscript, we demonstrate that our training approach can be applied to existing models (D3U, NsDiff), improving their inference efficiency.
>
> Table: Efficiency and performance comparisons on ETTh1.
>
> | Model | Pred. Len | MSE (std) | MAE (std) | CRPS (std) | CRPS_SUM (std) | Train T₁ (s/iter) | Train T₂ (s/iter) | Inference (min/batch) | NFEs | GPU Mem (MiB) |
> | --- | --- | --- | --- | --- | --- | --- | --- | --- | --- | --- |
> | SSOL | 336 | 0.491 (0.004) | 0.477 (0.003) | 0.458 (0.004) | 0.639 (0.006) | 0.023 | 0.027 | 0.022 | 1 | 364 |
> | D3U | 336 | 0.512 (0.006) | 0.478 (0.003) | 0.351 (0.002) | 0.922 (0.109) | 0.025 | — | 0.078 | 20 | 582 |
> | NsDiff | 336 | 0.728 (0.136) | 0.583 (0.046) | 0.431 (0.038) | 1.083 (0.171) | 0.105 | — | 0.544 | 20 | 2560 |
> | SSOL | 720 | 0.539 (0.014) | 0.535 (0.010) | 0.508 (0.013) | 0.672 (0.045) | 0.028 | 0.036 | 0.044 | 1 | 646 |
> | D3U | 720 | 0.533 (0.044) | 0.505 (0.022) | 0.371 (0.017) | 1.458 (0.862) | 0.030 | — | 0.164 | 20 | 884 |
> | NsDiff | 720 | 0.704 (0.062) | 0.613 (0.027) | 0.440 (0.020) | 2.086 (0.159) | 0.179 | — | 0.608 | 20 | 5380 |
>
> > Q6: Error bars for the imputation experiment in Fig 5.
> >
> Yes, you are correct that our 90% credible intervals are quite wide, especially at high missing ratios. This actually suggest that our probabilistic approach is working well for the following reasons:
>
> (a) These empirical quantile-based error bars (5th-95th percentiles from 100 samples) directly reflect the range of plausible values rather than standard error-based intervals. For a signal, when most of its values are missing (90% missingness), the conditional distribution given the limited observed points naturally becomes broad, and our probabilistic model shows this uncertainty.
>
> (b) Even when we have wide intervals, our median predictions (red dotted) tracks the ground truth well. This is very good and shows our model correctly identifies the most likely trajectory while also correctly representing uncertainty.
>
> (c) We find that the intervals appear to vary by signal characteristics: smoother channels (1 and 3) show narrower bands while other signals exhibit broader uncertainty, showing our model adapts its confidence.
>
> We fully agree that the blue dots for missing points are non-standard. We will revise and use solid dots for observed values, with imputed regions clearly marked.
>
> > Q7: Fig 3 (right): Why does performance improve with more missing data (10% to 25%)?
> >
> This behavior likely stems from how our model (and the FAB module) leverages global temporal patterns during training. At 10% missing ratio with geometric masking, the model sees many training examples with sparse, isolated gaps that must be filled using primarily local context (finer scale wavelet coefficients). This trains the model to rely heavily on immediate neighbors. At 25% missing ratio, the training data contains longer contiguous gaps that force the model to learn and exploit global patterns to bridge these gaps. Basically, the FAB module should adapt to prioritize low-frequency components. This results in a model that better captures the overall signal structure. When evaluated on the test set, this more globally-aware model (trained at 25%) can outperform the more locally-focused model (trained at 10%) because many time series, particularly Weather and Energy, have strong global patterns that, once learned, make imputation more accurate.
>
> > Q8: Error bars.
> >
>
> Yes, Table 1 lacks error bars. The point is well taken and we will add error bars to the table our SSOL. The gap with the baselines is large, so the benefits will be very clear in the table.
>
> For sampling stability, Table 6 reports standard deviations across inference runs, showing consistently small values. This confirms that our median-of-100-trajectories estimator produces statistically stable results. Table 5 shows, our method achieves better CRPS_sum performance on 5/6 datasets, confirming better joint distribution modeling. We agree that similar variability measures in Table 1 will reassure the reader. The additional results table shows performance across 3 random training initializations on extended horizons for ETTh1, with small standard deviations.
>
> > Q9: Place own contributions in context in related work.
> >
>
> We appreciate this constructive feedback. We will modify the text to clearly emphasize our contributions relative to existing approaches.
>
> > Q9: Suitability of the analysis in the spectral domain.
> >
>
> You are right that power spectral density (PSD) analysis must assume stationarity and discards phase information. We need to clarify two key aspects:
>
> (a) We do not operate on PSD: Figure 2 PSD visualization was simply a cartoon illustration of frequency-dependent smoothing. Our actual FAB module operates on complex-valued spectral coefficients that preserve both magnitude and phase. The inner product $\langle x_\tau, \chi_j \rangle$ retains full complex information.
>
> (b) Wavelet avoids stationarity problem: While our theory uses Fourier analysis for simplicity, our implementation uses wavelet transforms to avoid stationarity. Wavelets provide time-localized frequency analysis and our technical results carry through.
>
> The phase reconstruction issue you raised is relevant for Fourier-based approaches but does not apply to our wavelet implementation, where both approximation and detail coefficients are preserved throughout the forward and inverse transforms. We will revise Figure 2 and make sure that this confusion is avoided.

---

> ### Comment · Reviewer_bj9n · 2025-08-05
>
> I thank the authors for their responses to the points raised in my review. I am more satisfied with the clarifications provided now, however, I still find the explanations about computational complexity and error bars not fully convincing:
> - It's not only for 90% of missing data that the error bars are large (see fig 5 in the Appendix). As this is shown only for the proposed method, it is difficult to assess if these wide error bars are the appropriate ones (as mentioned by the authors' response)
> - In terms of computational complexity, from the table provided in the response, I see that the training time is slightly smaller for SSOL (vs D3U), in contrast to the paper's claim "significantly reducing computational overhead".
>
> Despite these flaws, I recognise the value of this work and will increase my score to recommend (marginal) acceptance.

---

> > ### Author Response · Authors · 2025-08-07
> >
> > We much appreciate your continued engagement, suggestions, and improvements to the paper. We carefully address the few remaining comments below.
> >
> > > It's not only for 90% of missing data that the error bars are large (see fig 5 in the Appendix). As this is shown only for the proposed method, it is difficult to assess if these wide error bars are the appropriate ones (as mentioned by the authors' response)
> > >
> >
> > We apologize for not explaining the error bar metric clearly enough. The wide bands in Figure 5 represent credible intervals showing the spread of possible imputed values (5th-95th percentiles across 100 generated samples at each time point). This addresses the question "what range of values are plausible for imputation given the missing data." But if the focus is on evaluating the precision of our estimator itself, i.e., "how accurately does the median/mean prediction estimate the true imputed value," then the appropriate metric would be the standard error $SE = \frac{\sigma}{\sqrt{100}}$, where $\sigma$ is the sample standard deviation of the 100 samples. The standard error-based confidence interval would be roughly 10 times narrower than our current credible intervals because it measures the precision of our aggregated estimator rather than the variability of individual samples. Both measures serve valid but different purposes: our current credible intervals characterize the inherent uncertainty in the imputation task, while standard error bars would characterize the reliability of our specific estimator.
> >
> > > In terms of computational complexity, from the table provided in the response, I see that the training time is slightly smaller for SSOL (vs D3U), in contrast to the paper's claim "significantly reducing computational overhead".
> > >
> >
> > Good point! Yes, we completely agree that our training time is comparable to D3U and apologize if our claim of "significantly reducing computational overhead" led to any misleading expectations. To clarify: our efficiency gains are primarily at **inference** time, not training. Our single-step generation achieves 3.5x speedup over D3U and 25x speedup over NsDiff, while training time depends on the backbone architecture. We will revise our claim to "reducing inference compute time" to avoid confusion and correctly identify where the main benefits are.

---

### Official Review · Reviewer_XQ6p · 2025-06-26

**Clarity:** 4
**Significance:** 3
**Originality:** 4
**Rating:** 5
**Confidence:** 4

**Summary:**

The paper introduces Single-Step Operator Learning (SSOL), a diffusion-based framework that reconstructs or forecasts multivariate time-series in a single denoising step. SSOL first models how the forward stochastic differential equation exponentially dampens high-frequency modes, then counteracts this effect with a learnable Frequency-Aware Block that projects each noisy sample onto a spectral basis and re-weights its coefficients as a function of the noise scale. Building on semigroup theory, the authors define an inverse operator $\phi$ that blends the spectral denoiser with the identity map and is explicitly trained to satisfy a Markov composition law, enabling a direct jump from the maximum-noise level to clean data without intermediate steps. A two-stage training objective aligns the model with the denoising boundary and progressively enforces the composition constraint over a refined noise grid. Experiments on six standard forecasting datasets and two imputation benchmarks show that SSOL matches or exceeds multi-step diffusion baselines while reducing function evaluations from 20 to 1, yielding roughly a twenty-fold speed-up.

**Questions:**

1.	Can you provide more insight or theoretical justification into when and why a single-step operator is sufficient for accurate approximation?
2.	Have you explored hybrid settings where part of the operator is learned via single-step and part via recurrence or iterative refinement?
3.	How does the architecture scale with grid resolution and dimensionality? Any observed bottlenecks in training or inference?
4.	Would your method generalize to non-grid or graph-based operator learning tasks?
5.	Could you release the code and data to support reproducibility?

**Ethical Concerns:**

["NO or VERY MINOR ethics concerns only"]

**Final Justification:**

The paper shall be acceptable based on all the clarification and new experiment results.  I rated the paper as 5.

**Limitations:**

The paper does not explicitly discuss limitations or societal impact. Please consider adding a short section on cases where single-step learning may underperform (e.g., highly nonlinear or chaotic systems).

**Paper Formatting Concerns:**

None. The paper complies with NeurIPS 2025 formatting.

**Quality:**

3

**Strengths And Weaknesses:**

Strengths:

1. The paper is technically solid and grounded in clear theoretical insights. The derivation of frequency damping in the forward SDE is rigorous, and the single-step inverse operator is carefully tied to semigroup composition, making the core idea both novel and well justified.

2. The proposed Frequency-Aware Block is explained in sufficient mathematical detail, and the accompanying algorithms are easy to follow.

3. Empirical validation is comprehensive: six forecasting datasets and two imputation tasks show that SSOL can match or surpass strong multi-step diffusion baselines while cutting the number of function evaluations from 20 to 1, yielding impressive practical speed-ups.

4. The manuscript itself is well structured, with clear sectioning, helpful figures, and a thorough reproducibility checklist, so readers can trace claims without undue effort.

Weaknesses:

1. Despite its merits, the study omits direct comparisons with the most recent single-step or consistency-distillation samplers (e.g., Consistency Models 2023), leaving readers unsure whether SSOL is competitive with the latest approaches in the broader diffusion literature.

2. The method’s reliance on a fixed spectral basis also raises questions about robustness: sensitivity to Fourier versus wavelet choices, and to irregular or non-uniform sampling, is not explored.

3. While the paper demonstrates good results up to a horizon of 192 steps, it remains unclear how well the approach scales to substantially longer horizons or to online, continually updated streams.

4. Finally, although the authors promise to release code, the current submission does not yet include a public repository or seeding scripts, which limits immediate reproducibility.

---

> ### Author Rebuttal · Authors · 2025-07-31
>
> We thank Reviewer XQ6p for the constructive feedback and thoughtful questions. We have carefully addressed all points below.
>
> > W1: Is SSOL competitive with the latest approaches in the broader diffusion literature?
> >
>
> While a number of methods have shown impressive single-step generation for images, adapting them to conditional time-series tasks requires substantial re-engineering that goes well beyond simple adjustments to obtain a baseline.
>
> In early stages of our work, the key adaptation challenges we encountered included sensible conditioning mechanisms, redesign of architectures and training objectives. For instance, image uses cross-attention with text/class embeddings. Time-series forecasting/imputation requires conditioning on irregular masks for missing data, or causal historical context, and partial observations. This is quite different from standard text prompts. Many image-focused models use U-Net with 2D convolutions operating on spatial features. Time series requires causal architectures that respect temporal ordering and handling of variable-length sequences with missing data. Image methods optimize perceptual quality, whereas time-series need probabilistic calibration. Again, these are not insurmountable challenges, but getting the architecture to work well and offer a strong baseline requires effort and experimentation. For example, with masked time-series data, defining valid trajectories between partially observed states and complete sequences needs domain-specific modifications. In our preliminary experiments, a direct application of Consistency Models to time-series yielded poor results due to these mismatches, which in some sense were the starting point of the formulation described here. For this reason, we focused our comparisons on recent time-series-specific methods (D3U, NsDiff) that already handle these domain challenges, providing more meaningful baselines for our community.
>
> > W2: Sensitivity to Fourier versus wavelet choices, and to irregular or non-uniform sampling.
> >
>
> To fully resolve this question, we performed a set of experiments comparing these bases on the ETTh1 dataset (prediction length 96), where wavelet transforms with db1 basis achieved better performance (MSE: 0.375, MAE: 0.396) compared to Fourier transforms (MSE: 0.391, MAE: 0.412). This performance difference aligns with what we would expect: Fourier analysis assumes global stationarity, and wavelets provide localized time-frequency analysis that better handles the non-stationary patterns and work a bit better.
>
> Regarding irregular and non-uniform sampling, yes we agree that standard wavelet transforms require uniformly spaced samples and rely on dyadic scale relationships. However, our method can handle irregular missing patterns well, as shown through imputation experiments with 10\%-90\% missing ratios using geometric masks that simulate real-world sensor failures. This addresses one class of irregularity we encounter in practice. But yes, for truly irregularly sampled time series where sampling times themselves are non-uniform, extending our framework would require incorporating more specialized transforms like non-uniform FFT.
>
> > W3: How well does the approach scale to substantially longer horizons or to online, continually updated streams?
> >
>
> Thanks for the question! To address this concern, we conducted additional experiments on longer prediction horizons (336 and 720 steps) using the ETTh1 dataset. The results demonstrate that our method maintains competitive performance at extended horizons: SSOL achieves comparable or better deterministic predictions with consistently better joint distribution modeling (CRPS-SUM improvements of 31\% at 336 steps and 54\% at 720 steps compared with D3U). For streaming applications, our single-step design offers computational advantages. Each prediction requires only one forward pass rather than 20+ iterative denoising steps, reducing inference time. This efficiency enables real-time forecasting as new observations arrive. The conditioning mechanism enables the direct incorporation of new data points into updated lookback windows, allowing for continuous prediction updates in streaming scenarios.
>
> Table: Forecasting performance for ETTh1.
>
> | Model | Pred. Len | MSE (std) | MAE (std) | CRPS (std) | CRPS_SUM (std) |
> | --- | --- | --- | --- | --- | --- |
> | SSOL | 336 | 0.491 (0.004) | 0.477 (0.003) | 0.458 (0.004) | 0.639 (0.006) |
> | D3U | 336 | 0.512 (0.006) | 0.478 (0.003) | 0.351 (0.002) | 0.922 (0.109) |
> | NsDiff | 336 | 0.728 (0.136) | 0.583 (0.046) | 0.431 (0.038) | 1.083 (0.171) |
> | SSOL | 720 | 0.539 (0.014) | 0.535 (0.010) | 0.508 (0.013) | 0.672 (0.045) |
> | D3U | 720 | 0.533 (0.044) | 0.505 (0.022) | 0.371 (0.017) | 1.458 (0.862) |
> | NsDiff | 720 | 0.704 (0.062) | 0.613 (0.027) | 0.440 (0.020) | 2.086 (0.159) |
>
> > Q1: Can you provide more insight or theoretical justification into when and why a single-step operator is sufficient for accurate approximation?
> >
>
> Our approach is based on the observation that the exponential smoothing of the data distribution's high-frequency modes during diffusion (Thm 3.2), can be effectively reversed by a model that understands the data's own frequency components. The FAB module is explicitly designed for this, learning to restore the fine-scale details in the data that correspond to the information lost in the distribution. The theoretical justification for why this single step is stable comes from the semigroup composition property. By training our operator $\phi$ to satisfy the consistency constraint $\phi(\rho,\gamma,\phi(\gamma,\tau,x)) = \phi(\rho,\tau,x)$, we are teaching it a direct mapping. This forces the model to learn the geometry of the globally consistent denoising trajectory, making the final one-shot reconstruction robust and well-behaved.
>
> But this is not universally applicable. We expect its effectiveness is likely to diminish under certain conditions. As discussed in Appendix D, we work with linear drift and non-degenerate Gaussian noise. The performance on far more complex, state-dependent, or variance-expanding diffusion schedules remains open, a single step may not be sufficient to capture such intricate dynamics. Our FAB is specialized for the patterns found in regularly sampled time-series data. For entirely irregularly sampled sequences or data like high-frequency audio, the construction would likely be less effective without non-trivial modifications.
>
> > Q2: Have you explored hybrid settings where part of the operator is learned via single-step and part via recurrence or iterative refinement?
> >
>
> Thank you for this excellent question! Such hybrid settings are feasible within our framework through straightforward modifications to the semigroup composition property. A coarse-to-fine hybrid can be implemented by training the operator $\phi(\gamma, \tau, \cdot)$ where $\gamma > 0$ (e.g., $\gamma = 10$, $\tau = \sigma_{\max} = 80$) rather than $\phi(0, \tau, \cdot)$. This modification to the boundary condition in our two-stage training objective (Section 4.3) allows the single-step operator to map from maximum noise level $\sigma_{\max}$ to an intermediate noise level $\sigma(\gamma)$, while preserving the semigroup composition law $S^*_{\gamma \to \tau}$ for the high-noise regime. The remaining denoising from $\sigma(\gamma)$ to clean data can utilize standard iterative diffusion samplers.
>
> > Q3: How does the architecture scale with grid resolution and dimensionality? Any observed bottlenecks in training or inference?
> >
>
> Thank you for the question about scalability. As we mentioned in our previous response regarding temporal resolution, we conducted additional experiments on longer prediction horizons (336 and 720 steps), demonstrating that our method maintains competitive performance with CRPS-SUM improvements of 31\% at 336 steps and 54\% at 720 steps compared with D3U. Regarding dimensionality, our experiments covered a range from 7 channels (ETT datasets) to 862 channels (Traffic dataset) with various temporal frequencies from 15-minute to hourly intervals. However, we observe that computational costs do increase with both longer sequences and more channels. The FAB module uses wavelet transforms with $O(d \log d)$ complexity for sequence length $d$. For channel scaling, we process each of $C$ channels independently in frequency space ($O(C \cdot d \log d)$ total).
>
> > Q4: Would your method generalize to non-grid or graph-based operator learning tasks?
> >
>
> Yes, the core principles are generalizable to non-grid domains like graphs, though it would require adapting some components. The fundamental idea of learning a single-step inverse operator by enforcing the semigroup composition property is not inherently tied to a grid since the diffusion process itself can be defined on other data types. The primary challenge will be adapting the Frequency-Aware Block (FAB), which is currently designed for time-series using Fourier or wavelet transforms. For a graph-based task, this component would need to be replaced with perhaps Graph Fourier Transform (work from 2000s and early 2010, first by Maggioni and layer by Hammond; both these papers are excellent starting points). While the principle of modulating spectral coefficients to reverse the diffusion process remains the same, this adaptation would be a non-trivial architectural modification.
>
> > Q5: Could you release the code and data to support reproducibility?
> >
>
> Yes, we will release the full codebase and documentation to reproduce all experiments. We take reproducibility seriously and know that the easiest way to encourage adoption is to release code. Our code repository will be publicly available concurrently with the paper. Thanks.
>
> > Q6: limitations or societal impact.
> >
>
> We apologize for the inconvenience. We moved them to Appendix D for space saving and will relocate them to the main body.

---

> > ### Comment · Reviewer_XQ6p · 2025-07-31
> > **Thanks**
> >
> > Thank you for addressing my concerns. The paper is acceptable as also supported by other reviewers.

---

> > > ### Author Response · Authors · 2025-08-01
> > >
> > > Thank you so much for appreciating our work and providing feedback through the review process!

---

### Official Review · Reviewer_uKCQ · 2025-06-30

**Clarity:** 3
**Significance:** 3
**Originality:** 3
**Rating:** 5
**Confidence:** 4

**Summary:**

This paper proposes Single-Step Operator Learning (SSOL), an approach to time-series diffusion models that enables efficient, one-step generation by leveraging frequency-domain insights. Unlike traditional diffusion models requiring multiple iterative denoising steps, SSOL learns an inverse operator that restores clean signals from noisy ones in a single pass, using a frequency-aware block and enforcing semigroup consistency. Extensive experiments show that SSOL matches or outperforms state-of-the-art probabilistic and deterministic models in forecasting and imputation tasks, while achieving up to 20× faster inference.

**Questions:**

please see the weakness above

**Ethical Concerns:**

["NO or VERY MINOR ethics concerns only"]

**Final Justification:**

I appreciate the author's response. My concerns have been addressed, I've raised the score to 5.

**Limitations:**

Yes

**Quality:**

3

**Strengths And Weaknesses:**

strengths:

1. The studies problem is important as well as challenging.

2. The idea of proposing semigroup consistency is noval, which avoids one-shot denoising from collapsing during inference or degenerating into a simple regression model. And the overall framework is techically sound.

3. The experiments shows the effectiveness of the proposed method, interms of the sampling efficiency and prediction&imputation performance.

weakness:

1.  Lack of detailed comparison between the proposed method and other one/few -step diffusion works
What is the difference between the proposed single-step operator and the one step diffusion works for time series?
In line 377-379,  the authors mentioned that
“Some of these approaches often rely heavily on teacher-student training. In contrast, our work leverages  classical principles Pazy [1983], Henry [1981] to define a single-step inverse operator.” However, that does not address the benefits of the proposed method.

2. Also there are other one-step diffusion methods like [1], it would be better to compare with them if applicable.
[1] One Step Diffusion via Shortcut Models

3. For the overall result table 1, to compare the probabilistic methods, why not reporting their performance in metrics like CRPS? Since for those methods, it is important to compare their probabilistic forcasting ability.

---

> ### Author Rebuttal · Authors · 2025-07-31
>
> We thank Reviewer uKCQ for the valuable feedback and insightful questions. We have carefully addressed each point below.
>
> > W1: What is the difference between the proposed single-step operator and the one-step diffusion that works for time series?
> >
>
> Thanks for pushing us to make this point explicit. The main distinction is in how single-step capability is achieved. Many existing approaches typically fall into three categories: (1) distillation methods like consistency models in the image domain need a pre-trained teacher model and iterative distillation to compress multi-step trajectories. (2) Post-hoc approximations like TSDiff will continue to maintain multi-step sampling but will use single-step guidance approximations within each iteration. But overall, 20+ NFEs are still needed. (3) Sequential sampling ideas like D3U and NsDiff reduce to roughly 20 steps through backbone architecture improvements. But still, they are iterative.
>
> Our approach differs in that we build single-step capability directly into the model architecture and training objective. We enforce the semigroup composition property $\phi(\rho,\gamma,\phi(\gamma,\tau,x)) = \phi(\rho,\tau,x)$ during training. So, the operator learns globally consistent transformations across all noise levels rather than just endpoint mappings. We can appreciate that this is not a post-training compression because the model does learn from scratch to perform accurate single-step denoising. This has several benefits. First, there is no dependency on pre-trained models or distillation. Second, we can achieve true single-step generation (NFE=1) versus 20+ steps. Finally, the training optimizes what we want directly, i.e., single-step accuracy rather than hoping that multi-step trajectories can be compressed well. The frequency-aware design helps us exploit time-series structure, unlike generic acceleration (Table 2 shows some of these benefits). We hope this clarifies the doubt.
>
> > W2: Compare with one-step diffusion models from the image domain, such as [1].
> >
>
> Thank you for suggesting this recent result. Shortcut Models achieves one-step generation by learning direct mappings from noise to data, bypassing intermediate diffusion states. While conceptually this is similar to pursuing single-step generation, our approach and their formulation differ a fair bit.
> Some shortcut models train a separate network to directly predict clean data from pure noise, using pre-trained diffusion model only for generating training pairs. This needs a fully-trained diffusion model as a starting point as well as additional distillation-like training of the shortcut network. Finally, one would need to balance between shortcut and diffusion paths during inference. In contrast, we integrate single-step capability by baking it into the formulation via semigroup composition properties. This is from the ground up: no pre-trained model needed.
>
> We also want to note that adapting Shortcut Models to conditional time-series tasks would require substantial modifications: handling partial observations through masking, incorporating temporal causality, and replacing image-centric architectures with time-series-appropriate designs. Our frequency-aware approach specifically leverages how diffusion affects temporal patterns (Theorem 3.2), achieving true single-step generation while maintaining time-series-specific inductive biases. A detailed empirical comparison would be valuable, but we hope that the reviewer will agree that it would essentially require developing a new time-series-specific variant of Shortcut Models. For example, adapting shortcut models[1] to conditioned time series generation requires domain-specific modifications beyond the core self-consistency framework. While the fundamental algebra of learning step-size conditioned shortcuts remains applicable, several adaptations are needed: (1) Instead of categorical labels, the model would condition on historical observations, missing value patterns, and forecast horizons through appropriate embedding schemes; (2) The spatial DiT transformer would be replaced with causal attention mechanisms or temporal backbones (like our FAB) designed for sequential dependencies; (3) Introducing mask-aware loss functions optimized only on future/missing indices. These modifications show that the direct application of shortcut models to time series is not straightforward and requires the adaptation approach we propose.
>
> [1] Frans, K., Hafner, D., Levine, S., & Abbeel, P. (2025). *One Step Diffusion via Shortcut Models*.
>
> > W3: Report probabilistic metrics.
> >
>
> Thanks for the question. We evaluated our method using CRPS and CRPS-sum metrics, with results in Table 5 (Appendix C). SSOL achieves the best CRPS-sum performance on 5 out of 6 datasets and competitive individual channel CRPS performance. This shows our method's strength in modeling joint distributions across channels. We apologize for not including these metrics in the main results table.
>
> We initially presented MSE/MAE to allow comparison with both probabilistic and deterministic baselines, as many recent time-series papers report only point estimates. However, we agree that probabilistic metrics are important. The CRPS results show an interesting pattern: while D3U excels at per-channel uncertainty quantification (best CRPS), SSOL better captures inter-channel dependencies (best CRPS-sum), likely due to our frequency-aware design that preserves correlation structures. If accepted, we are allowed one additional page of content. We will move these probabilistic metrics to the main results table in the revised version.

---

> > ### Comment · Reviewer_uKCQ · 2025-08-03
> >
> > Thanks for answering my question. My concerns have been addressed.

---

> > > ### Author Response · Authors · 2025-08-06
> > >
> > > Thank you so much for your thoughtful comments and valuable suggestions. We will incorporate the clarifications into the updated paper.

---

### Official Review · Reviewer_pJbC · 2025-07-03

**Clarity:** 2
**Significance:** 2
**Originality:** 2
**Rating:** 4
**Confidence:** 3

**Summary:**

This paper introduces an operator-learning approach for conditioned time series diffusion models, enabling efficient single-step generation by leveraging the frequency-domain characteristics of both the data and the diffusion process itself. The core insight is that the forward diffusion process induces a structured, frequency-dependent smoothing of the data's probability density function, which can be effectively reversed by a module operating in the frequency space. By setting up an operator learning task with frequency-aware building blocks that satisfy semi-group properties and exploit time series structure, the proposed method achieves forecasting and imputation results comparable or superior to many multi-step diffusion schemes, while significantly reducing inference costs.

**Questions:**

1. What is the actual training time for the two-stage training (boundary denoising and semigroup constraints) that requires iterative optimization?

2. If the noise range during the inference stage is significantly smaller than the training setting and the performance seems to have dropped noticeably, how can this be explained?

3. In imputation tasks with sparse historical data, how does SSOL leverage limited information to preserve temporal coherence? Does it rely excessively on prior distributions in such cases?

4.Can you explain why your method can ensure performance with single-step sampling? "avoids the iterative error propagation and hyperparameter tuning" seems insufficient.

**Ethical Concerns:**

["NO or VERY MINOR ethics concerns only"]

**Final Justification:**

I think the author's response addressed most of my questions, but for one point, I cannot further increase my score: based on many methods in the image domain, I believe transferring them to the time series prediction field is not difficult. The author's rebuttal mentioned three points: 1. Differences in data structure. Data distribution is implicitly learned, and structural differences can at most lead to performance loss, which can be fully compensated for with additional techniques. 2. Model architecture choice. As far as I know, DiT is currently widely applied and does not require any substantial modifications to the Transformer. 3. Conflict in evaluation systems. MSE can be used for training in both cases. In summary, I believe the author's avoidance of discussing existing methods in the image domain is a debatable behavior, but considering the overall quality of the paper is good, I maintain my score of 4.

**Limitations:**

Not found

**Paper Formatting Concerns:**

Not noticed

**Quality:**

3

**Strengths And Weaknesses:**

Strength:

1. SSOL reduces sampling steps from multistep to 1, achieving faster inference without sacrificing accuracy and showing efficient single-step generation ability.

2. The Frequency-Aware Block (FAB) explicitly restores high-frequency components attenuated by forward diffusion, using wavelet transforms to capture local temporal patterns. This design aligns with the inherent frequency structure of time-series data.

3. SSOL demonstrates consistent performance across diverse time-series domains, highlighting its adaptability to varying data characteristics.

Weakness:
1.The author's citation format seems to have a major issue; the reference entries are mixed directly into the main text, which makes reading very difficult.

2. The idea of reducing denoising steps has already been extensively researched in image generation.
Most of these methods can be directly utilized on time-series forecasting tasks. However, the paper lacks comparisons with these approaches, such as consistency sampling or flow-matching-based methods, among others.

3. Performance degrades when inference-time noise distributions (e.g., narrower σ ranges) differ from training, indicating limited robustness to distribution mismatches.

---

> ### Author Rebuttal · Authors · 2025-07-31
>
> We thank Reviewer pJbC for the detailed feedback and thoughtful comments. We have carefully addressed each point below.
> > W1: Citation format.
> >
> Thank you for the feedback. We made a mistake because of a conflict due to using cite/citet with natbib and other packages. We have made this change, and it is now numeric, thanks!
> > W2: The paper lacks comparisons with existing step-reduction methods from image generation.
> >
> Thank you for this excellent comment. We agree. Indeed, our construction builds upon single-step generation principles from image-domain methods. For example, the semigroup setup is conceptually quite similar to consistency models since both enforce compositional properties across noise levels.
>
> The reviewer will agree that adapting any shortcut method to a time series requires numerous domain-specific design choices:
>
> 1. Conditioning: Image methods use cross-attention with text/class embeddings. For time-series forecasting/imputation, we need to condition on (a) partial observations with irregular masks, (b) historical context with temporal ordering, and (c) missing data patterns. Our mask-aware conditioning is the result of many iterations of adjusting various ideas in image generation shortcuts, and turns out to be different from what works well in image domains.
> 2. Backbone: Many methods for image generation use U-Net, for time-series we need architectures that respect temporal causality. Again, conceptually there are many possible ways to do this, e.g., causal convolutions or masked attention, but many strategies do not directly work well. Separately, the FAB module exploits time-series structure by operating in frequency space. While we can start from various architectures, from image-generation shortcuts, finding the right combination that handles both causality and leverages time-series structure well, as well as gives good experimental results, is a bit involved.
> 3. Training Objectives: Image methods often use perceptual losses. Time series requires point-wise accuracy and probabilistic calibration (CRPS). Our two-stage training (Algorithm 1) balances these objectives without requiring a pre-trained teacher model.
>
> Overall, we completely agree with the idea that adapting any image shortcut method to a time series is conceptually straightforward. But the implementation requires rethinking nearly every component. Each design choice cascades where conditioning mechanisms affect architecture choices, which impacts training objectives, which interacts with frequency processing. We describe one coherent path through this design space. For example, adapting MeanFlow[1] to time series would need several domain-specific modifications while keeping the core operations (flow-matching identity and one-step update) unchanged: (1) redefining the average-velocity field for 1-D temporal space; (2) replacing categorical conditioning with structured history plus mask embeddings; (3) implementing a causal 1-D Transformer or our FAB backbone with continuous-time position encodings; (4) introducing mask-aware loss functions optimized only on future/missing indices.
>
> A direct comparison with consistency/flow-matching methods would require deriving entirely new time-series-specific architectures based on those principles and would need to create new methods rather than simple adaptations.
>
> [1] Geng, Z., Deng, M., Bai, X., Kolter, J. Z., & He, K. (2025). Mean Flows for One-step Generative Modeling.
> > W3: Performance drops when inference noise ranges differ from training distributions.
> >
> Thank you for bringing this up. This behavior is expected. Our single-step operator $\phi(\gamma,\tau,x)$ is trained to implement the inverse semigroup mapping from specific noise levels. During training with $\sigma \in [0.002, 80]$, the operator learns the full transformation $\phi(0, 80, x)$, from maximum noise to clean data, by enforcing the semigroup composition  $\phi(\rho,\gamma,\cdot) \circ \phi(\gamma,\tau,\cdot) = \phi(\rho,\tau,\cdot)$.
> But when $\sigma_{max}$ is reduced at inference time (e.g., to 10), we are essentially asking the operator to perform $\phi(0, 10, x)$, a transformation it can compose through intermediate steps but has not directly optimized for as a single jump. The forward diffusion at $\sigma=10$ hasn't sufficiently smoothed the data distribution (Theorem 3.2), so the prior is too close to the data manifold rather than being near-Gaussian. This violates the assumption that we start from a noise distribution with sufficient entropy for the reverse operator to shape successfully into the target distribution. Figure 4 (right panel) confirms this: performance remains stable for $\sigma_{max} \in [50,80]$ where the prior is sufficiently noisy, but degrades below this range. This shows our model  is performing diffusion-based generation rather than learning a shortcut that ignores noise.
>
> > Q1: Actual training time.
> >
>
> We provide a runtime analysis in the table below, showing actual training and inference times measured on the same hardware setup. Our two-stage training shows comparable efficiency to existing methods, with Stage 1 (boundary denoising) and Stage 2 (semigroup constraints) requiring similar per-iteration times to baseline single-stage training. The additional semigroup constraint stage adds minimal training overhead compared to standard denoising training, while our training GPU memory usage remains lower than baselines. Although we do not achieve a pure 20× speedup in wall-clock time due to backbone architecture differences across methods, our approach shows strong efficiency gains during inference.
>
> The key findings are as follows:
>
> (a) Inference speedup: SSOL achieves 3.5x faster inference than D3U and 25x faster than NsDiff.
>
> (b) Memory efficiency: 35% less GPU memory than D3U, 85% less than NsDiff.
>
> (c) Performance parity: Despite far fewer denoising steps, SSOL achieves comparable/better MSE/MAE.
>
> The efficiency gains come from our single-step design. The additional training time for our second stage (semigroup consistency) is negligible compared to the inference savings.
>
> Table: Runtime analysis for ETTh1.
>
> | Model | Pred. Len | MSE | CRPS_SUM | Train T₁ (s/iter) | Train T₂ (s/iter) | Inference (min/batch) | NFEs | GPU Mem (MiB) |
> | --- | --- | --- | --- | --- | --- | --- | --- | --- |
> | SSOL | 336 | 0.491 | 0.639 | 0.023 | 0.027 | 0.022 | 1 | 364 |
> | D3U | 336 | 0.512 | 0.922 | 0.025 | — | 0.078 | 20 | 582 |
> | NsDiff | 336 | 0.728 | 1.083 | 0.105 | — | 0.544 | 20 | 2560 |
> | SSOL | 720 | 0.539 | 0.672 | 0.028 | 0.036 | 0.044 | 1 | 646 |
> | D3U | 720 | 0.533 | 1.458 | 0.030 | — | 0.164 | 20 | 884 |
> | NsDiff | 720 | 0.704 | 2.086 | 0.179 | — | 0.608 | 20 | 5380 |
>
> > Q2: How does SSOL maintain temporal coherence with sparse historical data? Does it over-rely on priors in such cases?
> >
> Good question! Our SSOL method preserves temporal coherence through at least three key components/modules. First, the FAB module decomposes signals into multiple scales. So, even with sparse observations, low-frequency components can often be reliably estimated. This gives a sensible skeleton for reconstruction. High-frequency details can then be conditionally sampled consistent with this skeleton. Also, unlike unconditional generation that relies purely on priors, our operator $\phi$ is explicitly conditioned on both the mask $M$ and the observed values. During training, the model must figure out how to propagate information from observed regions through the semigroup structure. This allows learning how different masking patterns affect the conditional distribution. Finally, as observations become sparser, the model avoids blindly defaulting to generic priors. Instead, it adaptively widens its predictive distribution (Figure 5). The median estimator extracts the most likely trajectory while the widening credible intervals give some signal about when reconstruction is speculative versus data-driven. Rather than excessively relying on priors, we believe that SSOL learns observation-specific conditional distributions. With 90% missing data, the model must leverage learned temporal structures more heavily, but this is appropriate given the information constraints.
> > Q3: Explain why the proposed method can ensure performance with single-step sampling.
> >
> Our single-step performance guarantee stems from enforcing the mathematical semigroup composition property $S_{\gamma \to \tau}^* \circ S_{\rho \to \gamma}^* = S_{\rho \to \tau}^* $ during training. We learn an inverse operator $\phi(\gamma,\tau,\cdot)$ that must satisfy the semigroup composition constraint $\phi(\rho,\gamma,\phi(\gamma,\tau,x)) = \phi(\rho,\tau,x)$ for all $0 \leq \rho \leq \gamma \leq \tau$, meaning any noise transition should yield identical results whether taken as a direct jump $(\tau \to \rho)$ or decomposed through intermediate steps $(\tau \to \gamma \to \rho)$. Our two-stage training first uses boundary conditions via standard denoising loss (Stage 1), then explicitly encourages this composition property using an adaptive grid scheduler $N(\cdot)$ that progressively samples intermediate noise levels $\sigma_n = (\sigma_{\max}^{1/\upsilon} + \frac{n}{N(k)-1}(\sigma_{\min}^{1/\upsilon} - \sigma_{\max}^{1/\upsilon}))^\upsilon$ and minimizes $\mathcal{L}\_{cmp}=|\phi(\rho,\tau,x_\tau)-\phi(\rho,\gamma,\phi(\gamma,\tau,x_\tau))|\_2^2$ (Stage 2). This objective forces the operator to learn transformations that are consistent across all noise scales, while our frequency-aware block $B_\theta(x_\tau,\sigma(\tau)) = \sum_j \Psi_\theta(\alpha_\theta(\sigma(\tau)) \cdot \langle x_\tau,\chi_j \rangle)\chi_j$ specifically targets the exponential damping of high-frequency modes identified in Theorem 3.2. We are training the operator to approximate the semigroup composition, so $\phi(0,\tau,\cdot)$ learns to approximate inverse transformation $(S_{0 \to \tau}^*)^{-1}$ through semigroup property constraints rather iterative approximations.

---

> > ### Comment · Reviewer_pJbC · 2025-08-01
> >
> > Thanks for answering my question. I will keep my score.

---

> > > ### Author Response · Authors · 2025-08-06
> > >
> > > Thank you so much for your insightful feedback and constructive suggestions. The clarifications identified through our discussion will be reflected in the revised manuscript.

---

### Note · Authors · 2025-08-13

We sincerely thank all reviewers for their constructive feedback and thoughtful engagement throughout this review process. These productive discussions have allowed us to clarify our core contributions and address all of the concerns that were raised.

We are pleased that the reviewers recognized the key strengths of our work. Our single-step operator learning achieves generation in a single denoising step (NFE = 1) while maintaining or exceeding the performance of multi-step diffusion methods that typically require 20+ steps. The reviewers’ suggestions were very helpful, and we will incorporate both their recommendations and the clarifications provided in our rebuttals into the final manuscript. We are grateful for the time the reviewers invested in making our work stronger.

---

### Decision · Program_Chairs · 2025-09-17

**Decision:**

Accept (poster)

**Comment:**

This paper proposed a diffusion model for time series based on operator learning.  The approach reduces the generation problem from the usual multi-step diffusion methods to a single step. The authors developed their approach to mitigate the frequency-dependent smoothing induced by the forward operator of the diffusion by working in the frequency domain. The paper contains extensive experiments demonstrating the method's improved performance and computational complexity.

The reviewers highlighted a number of strengths
* The novel approach encouraging semi-group consistency to reduce the sampling steps to 1 makes inference faster without losing accuracy.
* The paper is technically solid with rigorous derivations and motivations.
* The comprehensive experimental evaluation was well done where the method was demonstrated to produce consistent predications across diverse domains.

The reviewers did bring up a few weaknesses related to comparing the method to existing work, some design choices, and evaluations on other relevant tasks (e.g. long-horizon prediction). All of these concerns were addressed.

One concern raised by a reviewer related to not comparing to existing working on single-step generation from the image generation literature. After substantial discussions between the reviewer and authors, these comparisons would require substantial changes to the existing methods at which point the authors would just be creating a (potentially sub-optimal) competitor to their proposed algorithm.

Despite this last concern, the proposed method and comprehensive evaluations make this a strong paper that should be accepted.